# Human neocortical expansion involves glutamatergic neuron diversification

The neocortex is disproportionately expanded in human compared with mouse[1,2], both in its total volume relative to subcortical structures and in the proportion occupied by supragranular layers composed of neurons that selectively make connections within the neocortex and with other telencephalic structures. Single-cell transcriptomic analyses of human and mouse neocortex show an increased diversity of glutamatergic neuron types in supragranular layers in human neocortex and pronounced gradients as a function of cortical depth[3]. Here, to probe the functional and anatomical correlates of this transcriptomic diversity, we developed a robust platform combining patch clamp recording, biocytin staining and single-cell RNA-sequencing (Patch-seq) to examine neurosurgically resected human tissues. We demonstrate a strong correspondence between morphological, physiological and transcriptomic phenotypes of five human glutamatergic supragranular neuron types. These were enriched in but not restricted to layers, with one type varying continuously in all phenotypes across layers 2 and 3. The deep portion of layer 3 contained highly distinctive cell types, two of which express a neurofilament protein that labels long-range projection neurons in primates that are selectively depleted in Alzheimer's disease[4,5]. Together, these results demonstrate the explanatory power of transcriptomic cell-type classification, provide a structural underpinning for increased complexity of cortical function in humans, and implicate discrete transcriptomic neuron types as selectively vulnerable in disease.

The neocortex is responsible for many aspects of cognitive function and is affected in numerous neurological and neuropsychiatric diseases. A prominent feature of the neocortex is its disproportionate expansion in surface area, volume and neuron number in large-brained mammals when compared to the expansion measured in subcortical structures[1,2]. Primate neocortex shows an increase in upper or supragranular layers[6], whose glutamatergic (excitatory pyramidal) neurons make connections to other neocortical and telencephalic brain regions[7].

This supragranular expansion, driven primarily by changes in gene regulation[8], includes increased cellular diversity. In rodents the pyramidal neurons of the supragranular neocortex (called layer 2/3 or L2/3 owing to lack of distinguishing boundaries) are relatively homogeneous within L2/3[7,9,10] and between cortical regions[7] (although there is variation in intracortical projection targets[11,12]). By contrast, primates show clear heterogeneity in layers 2 and 3 (L2 and L3) neurons in density, size, morphology, and electrophysiology as a function of cortical depth and projection target[1,13–20]. For example, h-channel (also called HCN channel) function shows marked variation by cortical depth, probably facilitating faithful transmission of signals for neurons with long apical dendrites[16]. Very large neurons in deeper L3 of non-human primates that express the non-phosphorylated form of heavy chain neurofilament protein and are immunoreactive to the antibody SMI-32 (SMI-32ir) preferentially send long-range corticocortical projections[21]. Macaque L3 neuron size, dendritic arborization, input resistance and firing patterns vary substantially by cortical region[11,22]. Two main human L3 pyramidal neuron types have been described that differ in dendritic morphology (slim- versus profuse-tufted[17]), and the basal dendritic arbor size of human neocortical pyramidal neurons increases from caudal to rostral neocortex[23,24]. Finally, the SMI-32ir neurons in deep L3 are preferentially vulnerable to

early degeneration and are dramatically reduced in late-stage Alzheimer's disease[4,5].

Single-cell or single-nucleus RNA sequencing (RNA-seq) provides a powerful strategy to quantitatively define neuronal diversity and compare cells across species[3,25–30], and demonstrates an increased diversity of supragranular glutamatergic intratelencephalic-projecting (IT) neurons in human compared to mouse[3]. Human L2 and L3 contain three abundant transcriptomic types (t-types), which map to the three t-types found in mouse L2/3[3], plus additional glutamatergic neuron types in deep L3 not found in mouse supragranular neocortex. We developed a robust Patch-seq[31,32] platform for human neocortical tissues from 90 neurosurgical resections to directly characterize the physiological and morphological properties of supragranular neurons and test whether the increased transcriptomic diversity of human glutamatergic IT types is mirrored in other cellular properties.

## Results

### Greater neuronal diversity in human neocortex

We used histology to compare the density and size of neurons from the human middle temporal gyrus (MTG), the most accessible region from neurosurgery, with two neocortical areas in mouse: the extensively characterized primary visual cortex (VISp)[25,26] and the temporal association area (TEa), which is often used as a comparator for MTG[16,19,33]. Human L2 and L3 exhibited significant differences in the size and density of neurons as a function of cortical depth compared with L2/3 in both mouse regions (Fig. 1a). The density of human neurons was graded (Fig. 1b, left), with the highest density in L2, decreasing by half to a minimum in mid L3 (Fig. 1b, right). By contrast, supragranular

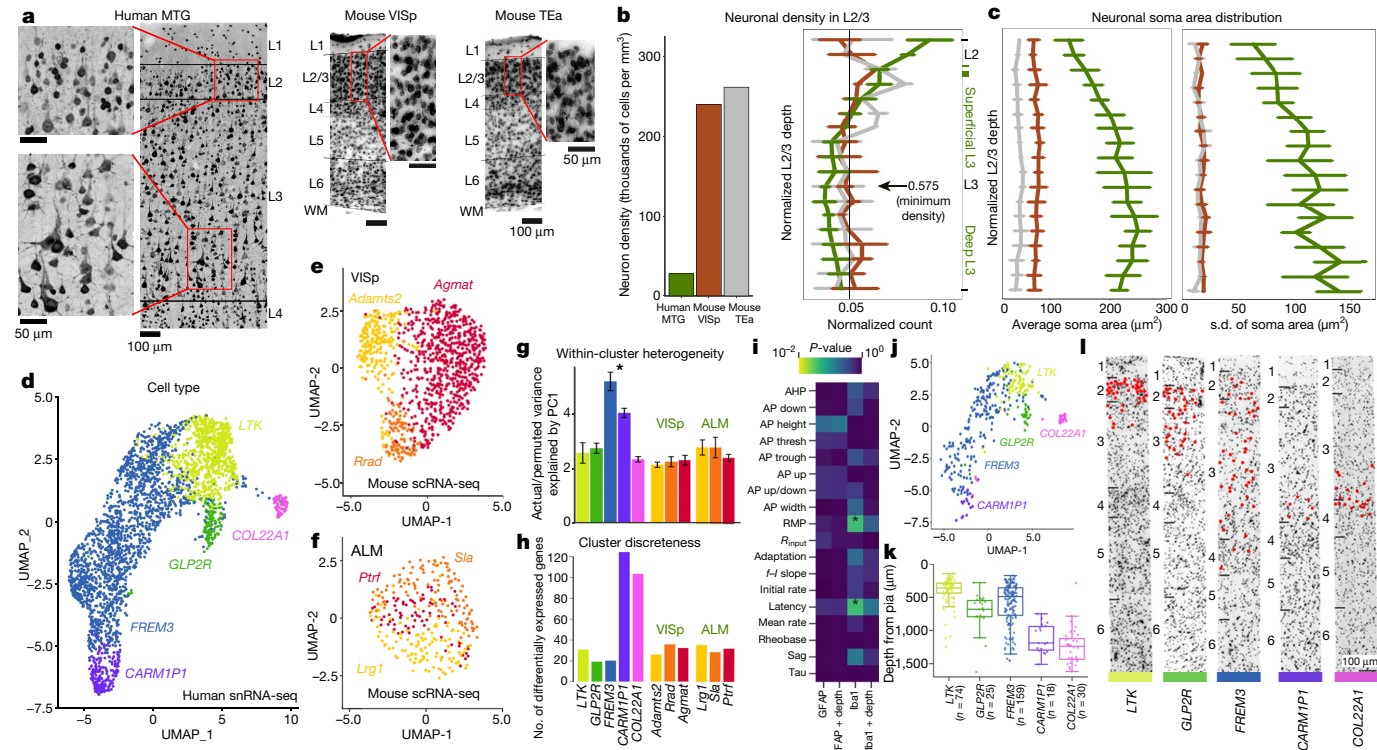

**Fig. 1 | Comparison of human versus mouse supragranular neurons. a**, NeuN labelling of neurons in human MTG (left), mouse VISp (center), and mouse TEa (right). Higher magnification insets spanning L2 and L3. WM, white matter. **b**, Left, neuron density through L2 and L3. Tick marks show individual donors. Right, Normalized histogram of neuron density in mouse VISp (red), mouse TEa (grey) and human (green) L2-3. Normalized L2/3 depth is defined as distance from L1–L2 boundary to soma/(L2/3 thickness). **c**, Mean (left) and standard deviation (right) of soma area. Green tick marks indicate border between L2 and L3 for each human sample. Data in **b**, **c**, are mean ± s.d. of metrics across donors. **d**, UMAP of 2,948 dissociated human nuclei collected[3] from five glutamatergic t-types in L2 and L3 of human MTG using the 2,000 most binary genes. Cells are colour-coded by t-type. **e**, **f**, Comparable UMAP of 981 mouse cells[26] mapping to three glutamatergic L2/3 neuron types in VISp (**e**) and 313 cells mapping to ALM (**f**). **g**, Average variance explained by principal component 1 (PC1) across 100 subsets of actual versus permuted data (Methods). Error bars show s.d. **h**, Average number of differentially expressed

genes between indicated cluster and other homologous t-types. **i**, Comparison of calculated electrophysiological features (*y*-axis) between recorded neurons in low (0–1) versus high (2–3) score bins for GFAP and IBA1, with and without including cell depth as a regressor (*x*-axis). *$P < 0.05$ (FDR-corrected). AHP, after hyperpolarization; AP, action potential features: down, downstroke velocity; thresh, voltage threshold; trough, nadir following action potential; up, upstroke velocity; RMP, resting membrane potential; $R_{input}$, input resistance; adaptation, spike frequency adaptation ratio; *f*–*I* slope, slope of the firing rate versus current curve; rate, firing rate; latency, delay between stimulus onset and first AP; tau, time constant (details in Methods 'Electrophysiology feature analysis'). **j**, UMAP of 385 glutamatergic Patch-seq neurons from supragranular cortex in human MTG, colour-coded by mapped t-type and plotted as in **d**. **k**, Depth of human Patch-seq neurons, by t-type. **l**, Location of t-types within the neocortex (red dots) demonstrated using mFISH. Black lines delineate layer boundaries. t-Type is indicated as in **k**.

layers of both mouse regions showed sixfold higher neuronal density than in the human regions, with homogeneous distribution by depth. Similarly, the average cross-sectional area of neuron somata doubled from L2 to deep L3 in human (with the largest exceeding 850 μm²) but was uniform in mouse L2/3 (Fig. 1c, left). Furthermore, variation in deep L3 soma size was fourfold higher in human versus mouse (Fig. 1c, right), which was clearly visible in human histological sections with large and small neurons comingling (Fig. 1a).

In a previous RNA-seq study[3], molecularly-defined glutamatergic t-types were also found to be more diverse, with five t-types in human supragranular MTG (with *LTK*, *GLP2R*, *FREM3*, *CARM1P1* and *COL22A1* types) versus three t-types each in mouse VISp and ALM. As reported[3], the predominant human t-type *FREM3* showed an extended gradient that, with more lenient clustering criteria, could be divided into sub-types that varied by depth (Extended Data Fig. 1a), whereas other t-types in human and mouse showed tighter clustering (Fig. 1d–f). Within-type heterogeneity was high in *FREM3*, and to a lesser extent in the *CARM1P1* type, with greater homogeneity seen for all other mouse and human t-types (Fig. 1g). The distinctness between clusters, measured as the number of differentially expressed genes between pairs of types, was

highest for the deep L3 *CARM1P1* and *COL22A1* types (Fig. 1h). While these species differences may be partially owing to areal variation, at minimum, they are consistent across two very different mouse brain areas.

## Patch-seq on human neurosurgical tissues

Previous studies[16,18,19,34–36] have shown that neurosurgically excised human neocortical tissues can be used for slice patch clamp studies within about 12 h of resection (and later[37]), and that human MTG t-types are consistently identified by single-nucleus RNA sequencing (snRNA-seq) in post mortem and living resected tissue. Building on this, we developed a robust technology platform for Patch-seq[31,32,38] on acute slice preparations from human neurosurgically resected neocortical tissues (Extended Data Fig. 1, Supplementary Table 1) to characterize the electrophysiological, morphological and transcriptomic properties of living human L2-3 neurons using standardized stimuli, biocytin filling and RNA-seq analysis.

To determine whether resected tissues are inherently pathological, we quantified trends in neuronal properties for six histological markers of pathology (Methods). Surprisingly, on a scale of 0 (neurotypical) to 3 (most pathological), most cases had average scores below

1.0 (Extended Data Fig. 1c). These markers had very low correlation with each other, with only Ki-67 (dividing cells) and Nissl (cellularity) being modestly correlated (Extended Data Fig. 1d). Among 18 features compared between low (0–1) and high (2–3) scored cases for GFAP and IBA1 (the only markers not skewed heavily towards 0, Extended Data Fig. 1c), only resting membrane potential ($P = 0.041$) and first-spike latency ($P = 0.042$) differed between IBA1 groups (Fig. 1i). However, no effects of pathology were significant when including neuron depth as a factor ($P > 0.17$ for all features), indicating that these differences can be explained by imbalanced sampling across cortical depth for the few cases marked as overtly pathological by IBA1. We also did not observe obvious relationships between electrophysiological features and pathology, age or sex (Extended Data Fig. 2).

Human Patch-seq cells were mapped to the MTG reference classification[3] and assigned t-types using the integration and label transfer classification workflow in Seurat (V3) (Methods)[39,40]. After alignment, Patch-seq cells intermixed with dissociated cells and nuclei in a low-dimensional uniform manifold approximation and projection (UMAP) space (Fig. 1j), and cells assigned to different t-types were generally colocalized in distinct locations in this space, indicating good agreement between platforms. T-types showed discrete sub-laminar distributions that were consistent between Patch-seq biocytin staining and cellular marker distributions by multiplex fluorescence in situ hybridization (mFISH) (Fig. 1k, l). These results indicate that Patch-seq data are consistent with reference transcriptomic classifications from dissociated nuclei, and that mapping is robust despite many potential sources of technical noise and uncontrolled variation in the human tissue samples.

A total of 385 neurons that passed transcriptomic data quality control mapped with higher confidence to the five supragranular human glutamatergic t-types than to any other neuron types. Most neurons in the dataset preserved sufficient labelling to determine the relative depth of the soma with respect to the pia and the border between L2 and L3. A majority of neurons ($n = 283$) also produced sufficiently complete recordings to calculate electrophysiological features. The subset of neurons ($n = 109$) with sufficient biocytin fills and intact apical dendrites were imaged at high resolution, then subsequently manually reconstructed.

## Diversity of human excitatory t-types

In the aggregate, properties of human t-types sampled by Patch-seq were consistent with previous reports of slice physiology recordings from L2-3 pyramidal neurons[16,17]. However, with the transcriptome as the basis for classification, t-types showed clear qualitative morphoelectric differences (Fig. 2, Supplementary Note). One of the most obvious differences between human t-types was dendrite size, which varied markedly across the large thickness of human supragranular neocortex for t-types found in different layers with apical dendrites that extend to L1 (Fig. 2a, Extended Data Fig. 4). T-types also varied in their relative apical versus basal lengths as well as both passive, single action potential and sustained firing properties (Fig. 2a, b, Supplementary Note), as visualized by UMAP projections of electrophysiological and morphological data (Fig. 2c, d). The *COL22A1* type shows strong separation, whereas the other types vary more continuously with some overlap. In particular, the morphoelectric properties of the *FREM3* type span the range of, and partially overlap with, *LTK*, *GLP2R* and *CARM1P1* types. A sparse principal component analysis (SPCA) projection of electrophysiological features (Fig. 2c, right) shows small groups of features that determine the two axes of greatest variability. The first ($x$) axis is dominated by features related to passive membrane properties, including membrane time constant, input resistance and rheobase. Variability along the second ($y$) axis shows differences in action potential shape. Similarly, an SPCA projection of morphological features (Fig. 2d, right) shows that features related to the total size of

the dendrites (length, volume and surface area) are most prominent ($x$-axis), whereas additional features of the apical dendrites (vertical extent and bias) further capture the distinctness of the *COL22A1* type and the continuum within the *FREM3* type ($y$-axis). Box plots of individual features from the SPCA space emphasize that most, but not all, pairs of t-types differ significantly ($P < 0.05$, false discovery rate (FDR)-corrected Mann–Whitney test).

Although comparisons of morphoelectric neuronal properties between human MTG and mouse neocortex are problematic given the difficulties in identifying homologous regions and the high variance of these properties across regions, it is nevertheless valuable to assess whether the homologous mouse t-types are similarly phenotypically differentiated from one another in any cortical region. We analysed L2/3 pyramidal neurons from mouse VISp using the same Patch-seq platform[41]. Included were 120 neurons with high-quality electrophysiology and transcriptome data mapping to the three mouse L2/3 glutamatergic t-types, and 60 neurons with data in all three modalities—these neurons had heterogeneous electrophysiological and morphological properties but were more like one another than the human t-types (Extended Data Figs. 5, 6, Supplementary Note).

## Variation of the FREM3 type by depth

Graded change in properties by depth is a prominent organizational principle for the *FREM3* t-type[3] (Fig. 1d), where a transcriptomic UMAP projection shows a strong relationship between soma depth and gene expression (Fig. 3a). Similarly, multiple electrophysiological and morphological features (apical and basal dendrites) vary continuously with depth (Fig. 3b, c) in agreement with other studies[16,17,19]. *FREM3* neurons span the full depth of L2-3 and send apical dendrites more than 1 mm to L1 (Fig. 1l, 3b). Although features such as apical dendrite height necessarily correlate with the distance from the soma to L1, many independent features were also strongly correlated, including the maximum length of basal dendrites (Fig. 3b) and soma radius (Extended Data Fig. 7), with 37 of 58 morphological features correlated overall (FDR < 0.05; Supplementary Table 2). Graded electrophysiological properties also exist (Fig. 3c), with 9 of 18 measured features significantly correlated with depth (FDR < 0.05; Supplementary Table 2). The strongest correlations were observed for an increase in sag and action potential upstroke/downstroke ratio with depth and a decrease in action potential latency at rheobase (Fig. 3c). We found that the expression of 790 genes was correlated with depth (FDR < 0.05). Gene ontology (GO) enrichment analysis on this gene set (Fig. 3d, Supplementary Table 2) predicts functional variation across graded neuronal phenotypes based on enrichment of genes associated with synaptic transmission, projection morphogenesis and cell migration ($P = 3.14 \times 10^{-13}$, $7.54 \times 10^{-12}$, $5.9 \times 10^{-9}$).

## Deep t-types are phenotypically distinct

Deep L3 consists of a highly diverse set of putative IT projection types, including *CARM1P1*, *COL22A1*, and deep *FREM3* (defined by the neuronal density minimum in L2-3) (Fig. 1b). Compared with superficial types, deep neurons have larger apical dendrites, reflecting deeper soma locations, but a surprising lack of apical dendrite in L1 (Fig. 4a, bottom). Their axons also innervate L4 to a greater extent than superficial t-types (Extended Data Fig. 8). They differ electrophysiologically from superficial types with higher sag, action potential upstroke/downstroke ratio and initial instantaneous firing rate. This latter feature, reflecting a vigorous response to the onset of stimulation with a burst of action potentials followed by strong firing rate adaptation, may correspond to bursting phenotypes observed in L3 neurons in non-human primates[22].

The increased transcriptomic distinctness of deep types (Fig. 1h) is also reflected in their electrophysiology and morphological features. While every feature that differentiates between superficial neuron types also differentiates between deep types, several features uniquely

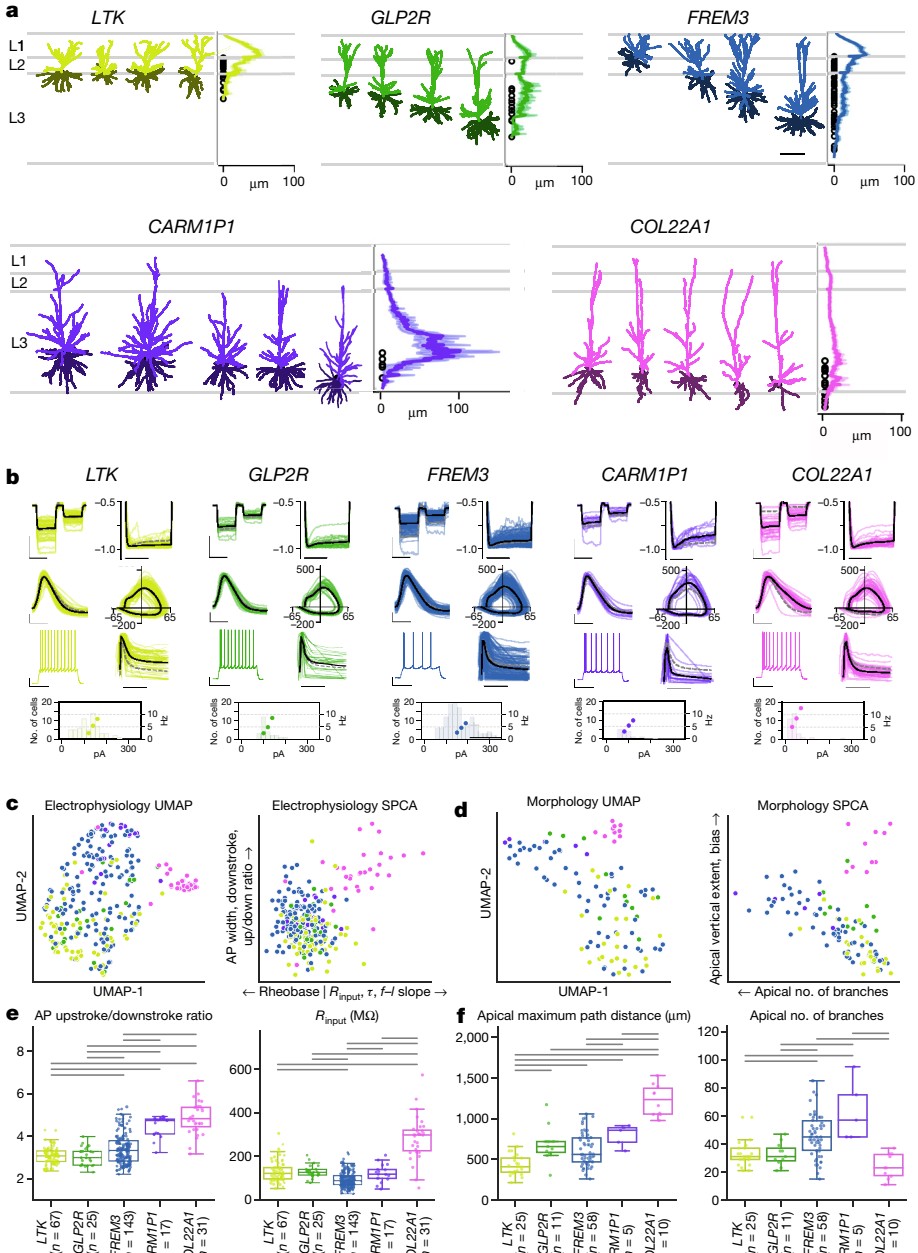

**Fig. 2 | Human L2 and L3 glutamatergic t-types show strong morphological and electrophysiological differentiation by t-type. a**, Morphology descriptions of the three prominent superficial (top row) and deep (bottom row) human L2 and L3 glutamatergic neuron t-types. For each t-type: left, representative examples of morphological reconstructions (scale bar, 250 μm); right, histogram of the average apical dendrite branch length by cortical depth and layer for all reconstructed cells from each t-type. **b**, Intrinsic electrophysiology responses by t-type. Coloured lines are individual neurons, solid black line represents the mean of all neurons in that t-type, dashed grey line is a global mean of the other t-types. Top row: left, responses to −70 and −30 pA current injections (scale bars, 10 mV, 1.0 s); right, responses to normalized to peak deflection to reveal voltage sag (scale bar, 0.5 s). Second row: left, first action potential during rheobase current injection (scale bar, 25 mV, 1.0 ms); right, corresponding phase plot (x-axis, mV; y-axis, mV ms⁻¹).

Third row: left, representative suprathreshold spiking response (scale bars, 20 mV, 0.5 s); right, normalized instantaneous firing rates for a suprathreshold pulse, demonstrating adaptation of firing rate (scale bar, 0.5 s). Bottom row: histogram of rheobase currents (left axis) and mean frequency to current curves (dots, right axis; currents normalized to the mean rheobase current before averaging). **c**, **d**, UMAP representation of electrophysiology (**c**) and morphology (**d**) space (left), and the same feature space projected onto sparse PCs (SPCA, right), with contributing features listed on each axis. **e**, **f**, Box plots showing feature distributions by t-type for illustrative features from each axis of SPCA space. Bars indicate significant pairwise comparisons ($P < 0.05$, FDR-corrected Mann–Whitney test). Boxes show median (centre line) and quartiles (top and bottom), whiskers show trimmed range bounded at 1.5× interquartile range beyond quartiles.

distinguish among deep types, including resting membrane potential, membrane time constant and action potential upstroke, as well as the width and complexity (for example, maximum branch order) of the basal dendrites (Extended Data Fig. 9). This distinctiveness is also seen in the

performance of a logistic regression classifier trained to predict t-type from electrophysiological features (Fig. 4b): superficial and deep types group separately, with higher discriminability among the deep types and the *FREM3* type forming an intermediate with similarity to both groups.

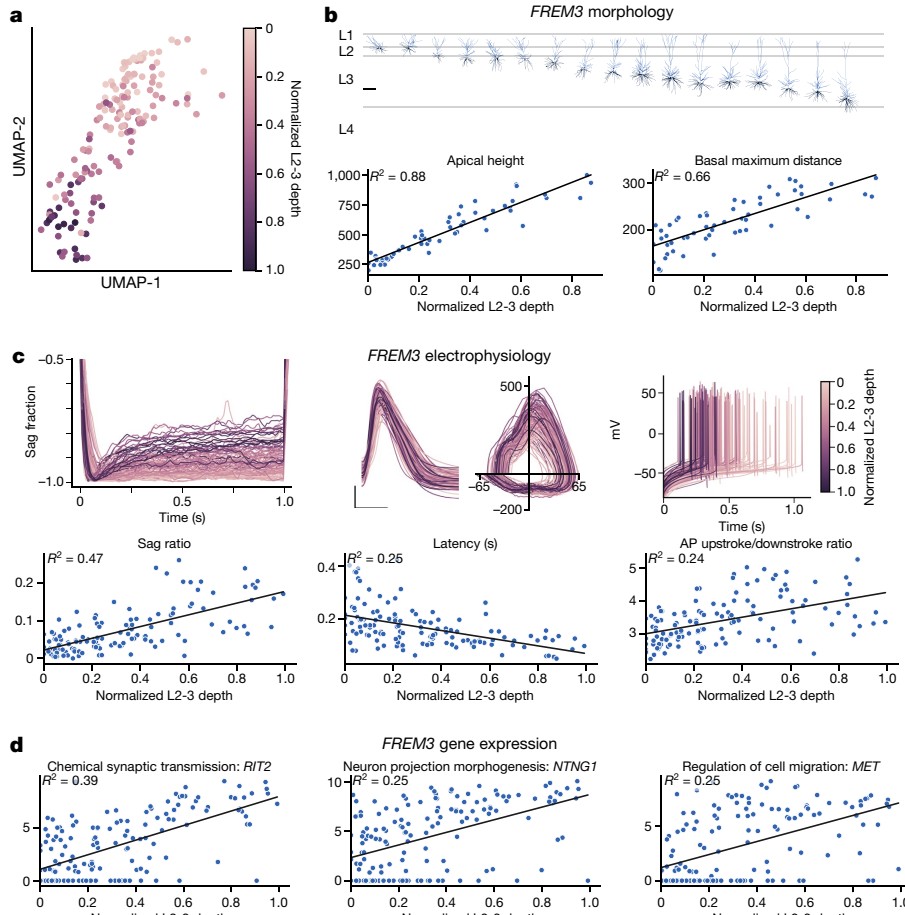

**Fig. 3 | Features of *FREM3* neurons vary according to laminar depth.**
**a**, *FREM3* neurons plotted in transcriptomic UMAP space (as in Fig. 1d). Each cell is coloured on the basis of its relative position within L2-3. Depth colour scale shown at right. **b**, Top, these *FREM3* neurons exhibit a range of morphologies spanning L2-3. Scale bar, 250 μm. Apical height and basal maximum distance are positively correlated with depth (bottom row). In **b**–**d**, all regressions shown are significant at FDR < $10^{-7}$. **c**, Top row, electrophysiology data traces coloured on the basis of each neuron's relative position within L2-3 (scale at right). Top left, hyperpolarizing pulses normalized to their peak deflection to allow for a sag comparison ($n = 124$). Top centre, overlaid first action potential during a rheobase current injection (scale bars, 25 mV, 1.0 ms; traces aligned to the time of threshold), as well as the corresponding phase plots (*x*-axis, mV; *y*-axis, mV ms$^{-1}$). Top right, initial action potentials at rheobase for 141 *FREM3* neurons aligned to the time of stimulus onset. Bottom, summary plots show that sag and action potential upstroke/downstroke ratio are positively correlated, and latency to action potential firing at rheobase is negatively correlated with depth. **d**, Representative gene examples for three GO categories with pronounced depth dependence of expression in *FREM3* neurons: chemical synaptic transmission, neuron projection morphogenesis and regulation of cell migration.

To understand transcriptional differences that may be predictive of phenotypic differences between *CARM1P1*, *COL22A1* and deep *FREM3* types, we used genesorteR[42] with slightly relaxed parameters (quant = 0.7) to identify genes selective for one or two of these three t-types and found 219 such marker genes (Extended Data Fig. 10). Differences in morphoelectric properties of the three deep L3 t-types were reflected in genes enriched for GO terms associated with neuronal connectivity, structure and synaptic signaling, including axon ($P = 3.5 \times 10^{-6}$; Bonferroni corrected), synapse ($P = 5.3 \times 10^{-5}$), calcium ion binding ($P = 0.008$) and extracellular matrix organization ($P = 0.00002$). Numerous genes involved in neuronal structure and function show specific expression in one or more deep types (Fig 4d), including those involved in dendritic branching (*COBLL1*[43]), neuron excitability (*KCNK2*[44]), long-term potentiation (*PHLDB2*[45]), and cannabinoid signaling (*CNR1*).

SMI-32 (encoded by the *NEFH* gene[21])-immunolabeled neurons preferentially make long-range ipsilateral projections[21] in L3 of the macaque temporal neocortex, and show selective vulnerability in Alzheimer's disease[4,5]. *NEFH* showed increased expression in deep *FREM3* and *CARM1P1* types relative to *COL22A1* and more superficial t-types

(Fig. 4d). Similarly, combined SMI-32 immunoreactivity and mFISH for cellular markers showed that the large *FREM3* and *CARM1P1* neurons were SMI-32-immunoreactive, whereas *COL22A1* neurons were not (Fig. 4e). This finding creates a putative link between transcriptomically-defined cell types, long-range projection target specificity and vulnerable neuron populations in Alzheimer's disease.

## Discussion

Human neocortical cellular diversity has been difficult to define quantitatively in part owing to underpowered analyses with low-throughput techniques because of limited tissue access, high variation across individuals and the absence of cell-type-selective tools. snRNA-seq can be applied to any species including human, where it forms the basis of a quantitative hierarchical cellular taxonomy that mirrors many aspects of cellular cytoarchitecture, function and developmental origins. Here we demonstrate using triple-modality Patch-seq analysis of human cortical neurosurgical resections that this transcriptomic classification is a Rosetta stone—that is, it is predictive of the morphological and electrophysiological diversity of supragranular neocortical neurons

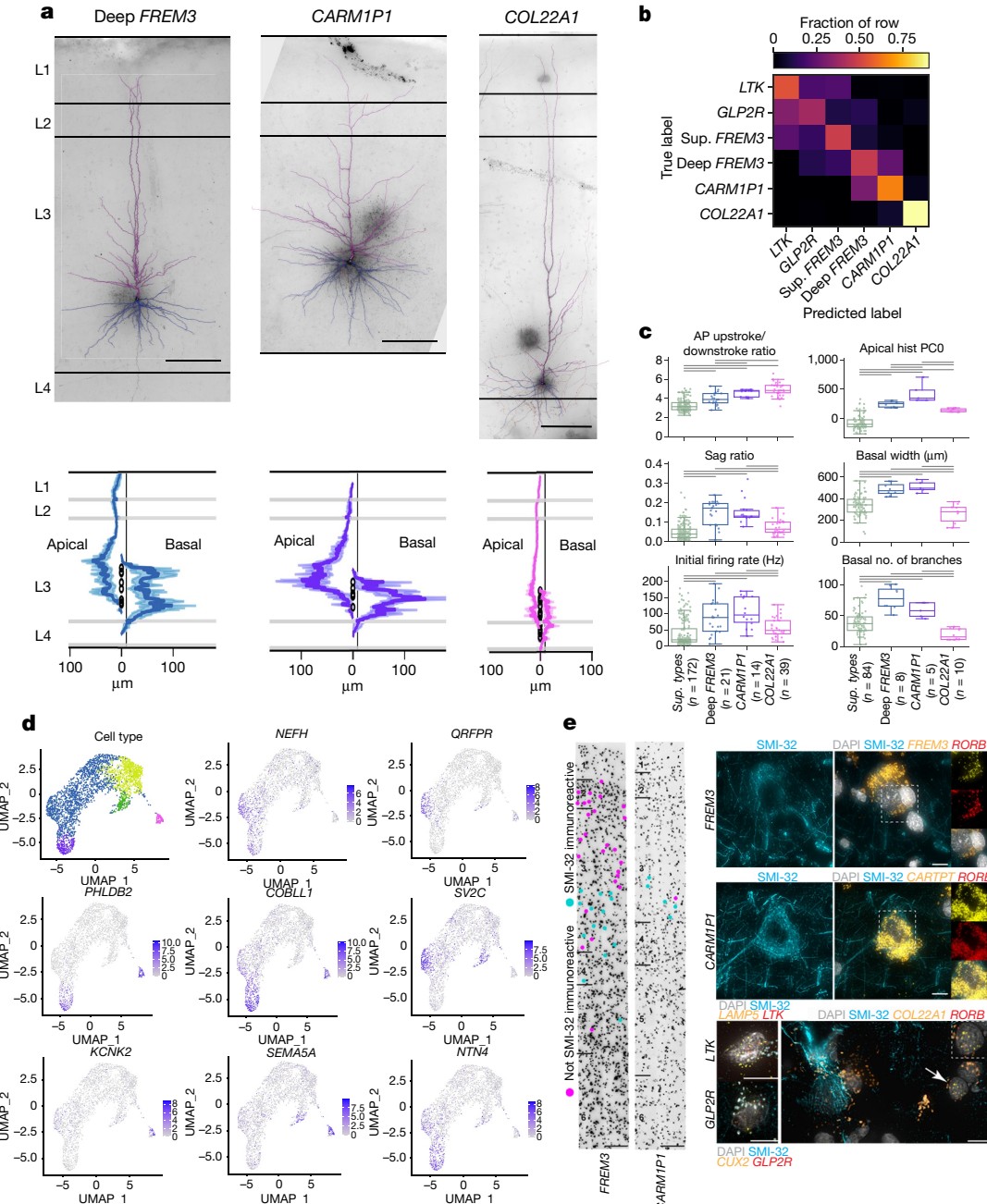

**Fig. 4 | Human deep L3 glutamatergic t-types are morphologically and electrophysiologically distinct. a**, Top, example reconstructions and associated maximum-intensity projection images of deep L3 t-types, deep *FREM3*, *CARM1P1* and *COL22A1* (scale bars, 200 μm). Bottom, histograms of the average apical and basal dendrite branch length (normalized to the maximum value for each t-type) by cortical depth and layer for all reconstructed cells from each t-type. Open circles indicate soma location. **b**, A logistic regression classifier predicts t-types on the basis of electrophysiological properties with 66% class-balanced accuracy for deep t-types, compared with 49% for superficial (sup.) t-types (overall 58% accuracy). **c**, Box plots of electrophysiology and morphology features that discriminate the three deep t-types from superficial t-types (*LTK, GLP2R* and superficial *FREM3*) and each other. Features shown selected from significant analysis of variance (ANOVA) results (FDR < $10^{-7}$ for electrophysiology, FDR < $10^{-2}$ for morphology). Bars indicate significant pairwise comparisons ($P$ < 0.05, FDR-corrected Mann–Whitney test). Boxes show median (centre line) and quartiles (top and bottom), whiskers show trimmed range bounded at 1.5× interquartile range beyond quartiles. Apical Hist PC0 is the first principal component of apical dendrite distribution with respect to cortical layer depths, representing a preference for apical found in L1 over deeper layers. **d**, A selection of eight marker genes that are differentially expressed in the deep L3 human t-types. Colour bars show normalized expression. The top left UMAP is identical to the one in Fig 1d. **e**, Left, MTG tissue immunostained for SMI-32. *FREM3* and *CARM1P1* neurons that are SMI-32 immunoreactive are indicated by cyan dots and those that are not are indicated by pink dots. Layer boundaries indicated at left of image, Scale bar, 100 μm. Representative SMI-32 immunoreactivity photomicrographs, along with mFISH for t-type specific genes shown for *FREM3* (top) and *CARM1P1* (middle) types. Right, representative mFISH composite images showing labelling for *DAPI, NEFH, RORB* and *FREM3* (top) or *CARTPT* (middle) in the same cell. The dashed box indicates the region of image shown on the right, where *RORB* and *FREM3* (top) or *CARTPT* (middle) are shown separately and then combined. Bottom, mFISH composite images with labelling for DAPI, neurofilament H, and t-type-specific genes for *LTK* (*LAMP5* and *LTK*), *GLP2R* (*CUX2* and *GLP2R*) and *COL22A1* (*COL22A1* and *RORB*) t-type. Scale bars, 10 μm. Marker gene expression is shown in Extended Data Fig. 3.

both for discrete and for continuous features. Specifically, we find the five glutamatergic t-types are distinct from one another, while the prominent transcriptomic gradient in the *FREM3* type is reflected in morphoelectric properties. Consistent with other studies using tissues distal to sites of obvious pathology, we find little evidence for disease- or pathology-related effects on electrophysiology or cell classification. This stands in contrast to studies that have used human tissues from pathological foci to characterize disease-related phenomena[46,47]. Remarkably, the stereotypy reported here holds true across 90 tissue donors, despite many uncontrolled axes of variation, indicating that the cellular blueprint is highly robust across individuals even in the context of disease.

Comparative analyses of transcriptome data from mouse, marmoset monkey and human strongly predicts that the human supragranular glutamatergic t-types belong to the IT subclass[3,28], and that the more superficially located *LTK*, *GLP2R* and *FREM3* types are homologous to the mouse supragranular IT types. By contrast, *CARM1P1* and *COL22A1* types do not have homologous types in mouse supragranular neocortex; rather, they are transcriptomically most like infragranular mouse IT types. Notably, the *COL22A1* type shares physiological (high input resistance and increased excitability) and morphological (simple, less branched dendrites including apical dendrites that often terminate before reaching L1) features with human L5 IT neurons[48]. This could reflect species differences in cell migration, or, more probably, an evolutionary co-option of a pre-existing IT transcriptional program and an extended developmental program. In either scenario, the outcome is increased neuronal diversity in human deep L3 that may have important functional implications. The enormous *CARM1P1* and deep *FREM3* neurons appear specialized for integration of local inputs. They both have complex, extensive (>0.5 mm) basal dendrites that are likely to integrate information across multiple adjacent minicolumns; by contrast, their apical dendrites sparsely enter layer 1, where feedback information is received from other cortical areas. Of note, the basal dendrites of these neurons undergo a substantial period of growth in early childhood when environmental factors have a crucial role in brain development[49]. *CARM1P1* and deep *FREM3* neurons are SMI-32-immunoreactive and *NEFH*-expressing, unlike *COL22A1* neurons, indicating that as in non-human primate they make long-range, predominantly ipsilateral projections compared with more locally projecting neurons. Increased local integration of deep L3 neurons supports an emerging hypothesis that in primates, superficial and deep parts of supragranular neocortex comprise functionally independent information streams, with feedforward projections originating in deep L3 neurons, whereas superficial neurons receive feedback information[50]. Finally, most deep L3 cells exhibit burst firing, a feature that optimizes information transfer[51]. Therefore, the increased cellular diversity in deep L3 may enhance efficiency of feedforward signal processing connecting distant regions of the expanded primate neocortex.

Finally, this cell classification may have potential for understanding the cellular locus of disease. SMI-32-immunoreative L3 magnopyramidal neurons are depleted in Alzheimer's disease progression[4,5], indicating selective vulnerability of the largest long-range association neurons and consequent disruption of cortical networks. Our results show that SMI-32 labelling maps onto the transcriptomic, morphological and physiological classification, labelling some large deep L3 types but not others. This refined morpho-electro-transcriptomic cellular framework may serve as a new roadmap for future studies investigating selective neuron disease vulnerability and resistance.

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

Jim Berg[1,14], Staci A. Sorensen[1,14], Jonathan T. Ting[1,2,14], Jeremy A. Miller[1,14], Thomas Chartrand[1], Anatoly Buchin[1], Trygve E. Bakken[1], Agata Budzillo[1], Nick Dee[1], Song-Lin Ding[1], Nathan W. Gouwens[1], Rebecca D. Hodge[1], Brian Kalmbach[1,2], Changkyu Lee[1], Brian R. Lee[1], Lauren Alfiler[1], Katherine Baker[1], Eliza Barkan[1], Allison Beller[3], Kyla Berry[1], Darren Bertagnolli[1], Kris Bickley[1], Jasmine Bomben[1], Thomas Braun[4], Krissy Brouner[1], Tamara Casper[1], Peter Chong[1], Kirsten Crichton[1], Rachel Dalley[1], Rebecca de Frates[1], Tsega Desta[1], Samuel Dingman Lee[1], Florence D'Orazi[1], Nadezhda Dotson[1], Tom Egdorf[1], Rachel Enstrom[1], Colin Farrell[1], David Feng[1], Olivia Fong[1], Szabina Furdan[5], Anna A. Galakhova[6], Clare Gamlin[1], Amanda Gary[1], Alexandra Glandon[1], Jeff Goldy[1], Melissa Gorham[1], Natalia A. Goriounova[6], Sergey Gratiy[1], Lucas Graybuck[1], Hong Gu[1], Kristen Hadley[1], Nathan Hansen[7], Tim S. Heistek[6], Alex M. Henry[1], Djai B. Heyer[6], DiJon Hill[1], Chris Hill[1], Madie Hupp[1], Tim Jarsky[1], Sara Kebede[1], Lisa Keene[3], Lisa Kim[1], Mean-Hwan Kim[1], Matthew Kroll[1], Caitlin Latimer[3], Boaz P. Levi[1], Katherine E. Link[1], Matthew Mallory[1], Rusty Mann[1], Desiree Marshall[3], Michelle Maxwell[1], Medea McGraw[1], Delissa McMillen[1], Erica Melief[3], Eline J. Mertens[6], Leona Mezei[5], Norbert Mihut[5], Stephanie Mok[1], Gabor Molnar[5], Alice Mukora[1], Lindsay Ng[1], Kiet Ngo[1], Philip R. Nicovich[1], Julie Nyhus[1], Gaspar Olah[5], Aaron Oldre[1], Victoria Omstead[1], Attila Ozsvar[5], Daniel Park[1], Hanchuan Peng[1], Trangthanh Pham[1], Christina A. Pom[1], Lydia Potekhina[1], Ramkumar Rajanbabu[1], Shea Ransford[1], David Reid[1], Christine Rimorin[1], Augustin Ruiz[1], David Sandman[1], Josef Sulc[1], Susan M. Sunkin[1], Aaron Szafer[1], Viktor Szemenyei[5], Elliot R. Thomsen[1], Michael Tieu[1], Amy Torkelson[1], Jessica Trinh[1], Herman Tung[1], Wayne Wakeman[1], Femke Waleboer[6], Katelyn Ward[1], René Wilbers[6], Grace Williams[1], Zizhen Yao[1], Jae-Geun Yoon[7], Costas Anastassiou[1], Anton Arkhipov[1], Pal Barzo[8], Amy Bernard[1], Charles Cobbs[7], Philip C. de Witt Hamer[9], Richard G. Ellenbogen[10], Luke Esposito[1], Manuel Ferreira[10], Ryder P. Gwinn[7], Michael J. Hawrylycz[1], Patrick R. Hof[11], Sander Idema[9], Allan R. Jones[1], C. Dirk Keene[3], Andrew L. Ko[10], Gabe J. Murphy[1,2], Lydia Ng[1], Jeffrey G. Ojemann[10], Anoop P. Patel[10], John W. Phillips[1], Daniel L. Silbergeld[10], Kimberly Smith[1], Bosiljka Tasic[1], Rafael Yuste[12], Idan Segev[13], Christiaan P. J. de Kock[6], Huibert D. Mansvelder[6], Gabor Tamas[5], Hongkui Zeng[1], Christof Koch[1] & Ed S. Lein[1,10 ✉]

[1]Allen Institute for Brain Science, Seattle, WA, USA. [2]Department of Physiology and Biophysics, University of Washington, Seattle, WA, USA. [3]Department of Pathology, University of Washington, Seattle, WA, USA. [4]byte physics, Berlin, Germany. [5]MTA-SZTE Research Group for Cortical Microcircuits, Department of Physiology, Anatomy, and Neuroscience, University of Szeged, Szeged, Hungary. [6]Department of Integrative Neurophysiology, Center for Neurogenomics and Cognitive Research (CNCR), Vrije Universiteit, Amsterdam, The Netherlands. [7]Swedish Neuroscience Institute, Seattle, WA, USA. [8]Department of Neurosurgery, University of Szeged, Szeged, Hungary. [9]Cancer Center Amsterdam, Brain Tumor Center, Department of Neurosurgery, Amsterdam UMC, Vrije Universiteit, Amsterdam, The Netherlands. [10]Department of Neurological Surgery, University of Washington, Seattle, WA, USA. [11]Nash Family Department of Neuroscience and Friedman Brain Institute, Icahn School of Medicine at Mount Sinai, New York, NY, USA. [12]NeuroTechnology Center, Columbia University, New York, NY, USA. [13]Edmond and Lily Safra Center for Brain Sciences and Department of Neurobiology, The Hebrew University Jerusalem, Jerusalem, Israel. [14]These authors contributed equally: Jim Berg, Staci A. Sorensen, Jonathan T. Ting, Jeremy A. Miller. ✉e-mail: Edl@alleninstitute.org

## Methods

Detailed descriptions of all experimental data collection methods in the form of technical white papers can also be found under 'Documentation' at http://celltypes.brain-map.org.

### Human tissue acquisition

Surgical specimens were obtained from local hospitals (Harborview Medical Center, Swedish Medical Center and University of Washington Medical Center) in collaboration with local neurosurgeons. All patients (Supplementary Table 1) provided informed consent and experimental procedures were approved by hospital institute review boards before commencing the study. Tissue was placed in slicing artificial cerebral spinal fluid (ACSF) as soon as possible following resection. Slicing ACSF comprised[52] (in mM): 92 N-methyl-D-glucamine chloride (NMDG-Cl), 2.5 KCl, 1.2 NaH$_2$PO$_4$, 30 NaHCO$_3$, 20 4-(2-hydroxyethyl)-1-piperazineethanesulfonic acid (HEPES), 25 D-glucose, 2 thiourea, 5 sodium-L-ascorbate, 3 sodium pyruvate, 0.5 CaCl$_2$.4H$_2$O and 10 MgSO$_4$.7H$_2$O. Before use, the solution was equilibrated with 95% O$_2$, 5% CO$_2$ and the pH was adjusted to 7.3 by addition of 5N HCl solution. Osmolality was verified to be between 295–305 mOsm kg$^{-1}$. Human surgical tissue specimens were immediately transported (15–35 min) from the hospital site to the laboratory for futher processing.

### Mouse breeding and husbandry

All procedures were carried out in accordance with the Institutional Animal Care and Use Committee at the Allen Institute for Brain Science. Animals (<5 mice per cage) were provided food and water ad libitum and were maintained on a regular 12-h light:dark cycle; rooms were kept at 21.1 °C and 45–70% humidity. Mice were maintained on the C57BL/6J background, and newly received or generated transgenic lines were backcrossed to C57BL/6J. Experimental animals were heterozygous for the recombinase transgenes and the reporter transgenes.

### Tissue processing

For mouse experiments, male and females were used between the ages of postnatal day (P)45 and P70 were anaesthetized with 5% isoflurane and intracardially perfused with 25 or 50 ml of 0–4 °C slicing ACSF. Human or mouse acute brain slices (350 μm) were prepared with a Compresstome VF-300 (Precisionary Instruments) or VT1200S (Leica Biosystems) vibrating microtome modified for block-face image acquisition (Mako G125B PoE camera with custom integrated software) before each section to aid in registration to the common reference atlas. Brains or tissue blocks were mounted for slicing with the optimal orientation for preserving intactness of apical dendrites of neocortical pyramidal neurons.

Slices were transferred to an oxygenated and warmed (34 °C) slicing ACSF for 10 min, then transferred to room temperature holding ACSF of the composition[52] (in mM): 92 NaCl, 2.5 KCl, 1.2 NaH$_2$PO$_4$, 30 NaHCO$_3$, 20 HEPES, 25 D-glucose, 2 thiourea, 5 sodium-L-ascorbate, 3 sodium pyruvate, 2 CaCl$_2$.4H$_2$O and 2 MgSO$_4$.7H$_2$O for the remainder of the day until transferred for patch clamp recordings. Before use, the solution was equilibrated with 95% O$_2$, 5% CO$_2$ and the pH was adjusted to 7.3 using NaOH. Osmolality was verified to be between 295–305 mOsm kg$^{-1}$.

### Patch clamp recording

Slices were bathed in warm (32–34 °C) recording ACSF containing the following (in mM): 126 NaCl, 2.5 KCl, 1.25 NaH$_2$PO$_4$, 26 NaHCO$_3$, 12.5 D-glucose, 2 CaCl$_2$.4H$_2$O and 2 MgSO$_4$.7H$_2$O (pH 7.3), continuously bubbled with 95% O$_2$ and 5% CO$_2$. The bath solution contained blockers of fast glutamatergic (1 mM kynurenic acid) and GABAergic synaptic transmission (0.1 mM picrotoxin). Thick-walled borosilicate glass (Warner Instruments, G150F-3) electrodes were manufactured (Narishige PC-10) with a resistance of 4–5 MΩ. Before recording, the electrodes were filled with ~1.0–1.5 μl of internal solution with biocytin (110 mM potassium gluconate, 10.0 mM HEPES, 0.2 mM ethylene glycol-bis(2-aminoethylether)-N,N,N',N'-tetraacetic acid, 4 mM potassium chloride, 0.3 mM guanosine 5'-triphosphate sodium salt hydrate, 10 mM phosphocreatine disodium salt hydrate, 1 mM adenosine 5'-triphosphate magnesium salt, 20 μg ml$^{-1}$ glycogen, 0.5 U μl$^{-1}$ RNAse inhibitor (Takara, 2313A) and 0.5% biocytin (Sigma B4261), pH 7.3). The pipette was mounted on a Multiclamp 700B amplifier headstage (Molecular Devices) fixed to a micromanipulator (PatchStar, Scientifica).

The composition of bath and internal solution as well as preparation methods were made to maximize the tissue quality, to align with solution compositions typically used in the field (to maximize the chance of comparison to previous studies), and modified to reduce RNAse activity and ensure maximal recovery of mRNA content.

Electrophysiology signals were recorded using an ITC-18 Data Acquisition Interface (HEKA). Commands were generated, signals were processed and amplifier metadata were acquired using MIES (https://github.com/AllenInstitute/MIES/), written in Igor Pro (Wavemetrics). Data were filtered (Bessel) at 10 kHz and digitized at 50 kHz. Data were reported uncorrected for the measured (Neher 1992) −14 mV liquid junction potential between the electrode and bath solutions.

Before data collection, all surfaces, equipment and materials were thoroughly cleaned in the following manner: a wipe down with DNA away (Thermo Scientific), RNAse Zap (Sigma-Aldrich) and finally with nuclease-free water.

For human slices, pyramidal shaped neurons in L2-3 were targeted. For mouse experiments, pyramidal neurons in L2/3 were targeted, either tdTomato$^-$ pyramidal neurons when recording from a transgenic line that labels interneurons, or tdTomato$^+$ neurons when recording from a line that labels different populations of L2/3 glutamatergic neurons, specifically Oxtr-T2A-Cre and Penk-IRES2-Cre-neo, each crossed to the Ai14 tsTomato reporter line.

After formation of a stable seal and break-in, the resting membrane potential of the neuron was recorded (typically within the first minute). A bias current was injected, either manually or automatically using algorithms within the MIES data acquisition package, for the remainder of the experiment to maintain that initial resting membrane potential. Bias currents remained stable for a minimum of 1 s before each stimulus current injection.

To be included in analysis, a cell needed to have a >1 GΩ seal recorded before break-in and an initial access resistance <20 MΩ and <15% of the input resistance ($R_{input}$). To stay below this access resistance cut-off, cells with a low input resistance were successfully targeted with larger electrodes. For an individual sweep to be included, the following criteria were applied: (1) the bridge balance was <20 MΩ and <15% of the $R_{input}$; (2) bias (leak) current 0 ± 100 pA; and (3) root mean square noise measurements in a short window (1.5 ms, to gauge high frequency noise) and longer window (500 ms, to measure patch instability) <0.07 mV and 0.5 mV, respectively.

Each cell was recorded using a standardized stimulus paradigm, including square pulses, ramps and noisy current injections, with the goal of extracting features that could be compared across cells, rather than tailoring each stimulus to the physiological input of that neuron.

Upon completion of electrophysiological examination, the pipette was centered on the soma or placed near the nucleus (if visible). A small amount of negative pressure was applied (~−30 mbar) to begin cytosol extraction and attract the nucleus to the tip of pipette. After approximately one minute, the soma had visibly shrunk and/or the nucleus was near the tip of the pipette. While maintaining the negative pressure, the pipette was slowly retracted in the x and z direction. Slow, continuous movement was maintained while monitoring pipette seal. Once the pipette seal reached >1 GΩ and the nucleus was visible on the tip of the pipette, the speed was increased to remove the pipette from the slice. The pipette containing internal solution, cytosol and nucleus was removed from pipette holder and contents were expelled into a PCR tube containing the lysis buffer (Takara, 634894).

### cDNA amplification and library construction

We performed all steps of RNA-processing and sequencing as described for mouse Patch-seq cells[41]. We used the SMART-Seq v4 Ultra Low Input RNA Kit for Sequencing (Takara, 634894) to reverse transcribe poly(A) RNA and amplify full-length cDNA according to the manufacturer's instructions. We performed reverse transcription and cDNA amplification for 20 PCR cycles in 0.65 ml tubes, in sets of 88 tubes at a time. At least 1 control 8-strip was used per amplification set, which contained 4 wells without cells and 4 wells with 10 pg control RNA. Control RNA was either Universal Human RNA (UHR) (Takara 636538) or control RNA provided in the SMART-Seq v4 kit. All samples proceeded through Nextera XT DNA Library Preparation (Illumina FC-131-1096) using either Nextera XT Index Kit V2 Sets A-D (FC-131-2001,2002,2003,2004) or custom dual-indexes provided by Integrated DNA Technologies (IDT). Nextera XT DNA Library prep was performed according to manufacturer's instructions except that the volumes of all reagents including cDNA input were decreased to 0.2× by volume. Each sample was sequenced to approximately 1 million reads.

### RNA-seq data processing

Fifty-base-pair paired-end reads were aligned to GRCh38.p2 using a RefSeq annotation gff file retrieved from NCBI on 11 December 2015 for human and to GRCm38 (mm10) using a RefSeq annotation gff file retrieved from NCBI on 18 January 2016 for mouse (https://www.ncbi.nlm.nih.gov/genome/annotation_euk/all/). Sequence alignment was performed using STAR v2.5.3[53] in two pass Mode. PCR duplicates were masked and removed using STAR option bamRemoveDuplicates. Only uniquely aligned reads were used for gene quantification. Gene counts were computed using the R Genomic Alignments package summarizeOverlaps function using IntersectionNotEmpty mode for exonic and intronic regions separately[54]. Expression levels were calculated as counts of exonic plus intronic reads. For most analyses, $\log_2$(counts per million (CPM) + 1)-transformed values were used.

### Morphological reconstruction

**Biocytin histology.** A horseradish peroxidase (HRP) enzyme reaction using diaminobenzidine (DAB) as the chromogen was used to visualize the filled cells after electrophysiological recording, and 4,6-diamidino-2-phenylindole (DAPI) stain was used to identify cortical layers as described previously[9].

**Imaging of biocytin-labelled neurons.** Mounted sections were imaged as described previously[9]. In brief, operators captured images on an upright AxioImager Z2 microscope (Zeiss, Germany) equipped with an Axiocam 506 monochrome camera and 0.63× Optivar lens. Two-dimensional tiled overview images were captured with a 20× objective lens (Zeiss Plan-NEOFLUAR 20×/0.5) in bright-field transmission and fluorescence channels. Tiled image stacks of individual cells were acquired at higher resolution in the transmission channel only for the purpose of automated and manual reconstruction. Light was transmitted using an oil-immersion condenser (1.4 NA). High-resolution stacks were captured with a 63× objective lens (Zeiss Plan-Apochromat 63×/1.4 Oil or Zeiss LD LCI Plan-Apochromat 63x/1.2 Imm Corr) at an interval of 0.28 μm (1.4 NA objective) or 0.44 μm (1.2 NA objective) along the z axis. Tiled images were stitched in ZEN software and exported as single-plane TIFF files.

**Morphological reconstruction.** Reconstructions of the dendrites and the full axon were generated for a subset of neurons with good quality transcriptomics, electrophysiology and biocytin fill. Reconstructions were generated based on a 3D image stack that was run through a Vaa3D-based image processing and reconstruction pipeline[55]. Images were used to generate an automated reconstruction of the neuron using TReMAP[56]. Alternatively, initial reconstructions were created manually using the reconstruction software PyKNOSSOS (Ariadne-service) or the citizen neuroscience game[57] Mozak (Mosak.science). Automated or manually-initiated reconstructions were then extensively manually corrected and curated using a range of tools (for example, virtual finger and polyline) in the Mozak extension (Zoran Popovic, Center for Game Science, University of Washington) of Terafly tools[58,59] in Vaa3D. Every attempt was made to generate a completely connected neuronal structure while remaining faithful to image data. If axonal processes could not be traced back to the main structure of the neuron, they were left unconnected.

Before morphological feature analysis, reconstructed neuronal morphologies were expanded in the dimension perpendicular to the cut surface to correct for shrinkage[17,60] after tissue processing. The amount of shrinkage was calculated by comparing the distance of the soma to the cut surface during recording and after fixation and reconstruction. A tilt angle correction was also performed based on the estimated difference (via CCF registration) between the slicing angle and the direct pia-white matter direction at the cell's location[9].

### Slice immunohistochemistry

**Immunohistochemistry.** Tissue slices (350 μm-thick) designated for histological profiling were fixed for 2–4 days in 4% paraformaldehyde (PFA) in phosphate-buffered saline (PBS) at 4 °C and transferred to PBS, 0.1% sodium azide for storage at 4 °C. Slices were then cryoprotected in 30% sucrose, frozen and re-sectioned at 30 μm using a sliding microtome (Leica SM2000R). Sections were stored in PBS with azide at 4 °C in preparation for immunohistochemical and Nissl staining. Specific probes (vendor, dilution) used were: Neu-N (Millipore, MAB377, 1:2,000); SMI-32 (Biolegend, 801704, 1:2,000); GFAP (Millipore, MAB360, 1:1,500); parvalbumin (Swant, PV235, 1:2,000); IBA1 (Wako 019-19741, 1:1,000); Ki-67 (Dako M724001-2, 1:200). Full immunohistology protocol details available at http://help.brain-map.org/download/attachments/8323525/CellTypes_Morph_Overview.pdf?version=4&modificationDate=1528310097913&api=v2

**Slide imaging.** Colorimetric immunohistochemistry and other histologically-stained whole slides (that is, Nissl-stained preparations) for bright-field imaging were scanned using an Aperio ScanScope XT slide scanner (Leica Biosystems). The samples were illuminated using a 21DC Halogen Lamp (Techniquip). Bright-field images were acquired using ScanScope Console (v101.0.0.18) and controller (ve101.0.4.446) at 10× magnification (objective lens 20×/0.75 NA Plan Apo, 0.5× magnifier) resulting in a pixel size of 1.0 μm per pixel.

**Pathology scoring.** For every case, each set of images per histological marker were independently scored by three pathologists using a 4-point scale, where 0 is normal and 3 is overtly pathological. Well-established histological markers were used to evaluate cellularity (Nissl), neuronal density and layer orientation (NeuN), astrogliosis (GFAP), microglial activation state (IBA1), non-phosphorylated neurofilament-H (using antibody SMI-32), and cellular proliferation (Ki-67).

### mFISH

Fresh-frozen human postmortem brain tissues were sectioned at 14–16 μm onto Superfrost Plus glass slides (Fisher Scientific). Sections were dried for 20 min at −20 °C and then vacuum sealed and stored at −80 °C until use. The RNAscope multiplex fluorescent v1 kit was used per the manufacturer's instructions for fresh-frozen tissue sections (ACD Bio), except that fixation was performed for 60 min in 4% paraformaldehyde in 1× PBS at 4 °C and protease treatment was shortened to 10 min. For combined RNAscope and immunohistochemistry, primary antibodies were applied to tissues after completion of mFISH staining. Primary mouse anti-neurofilament H (SMI-32, Biolegend, 801701) was applied to tissue sections at a dilution of 1:250. Secondary antibodies (1:250) were goat anti-mouse IgG (H+L) Alexa Fluor conjugates

(594 or 647, ThermoFisher Scientific A-11005 or 21235). Sections were imaged using a 60× oil-immersion lens on a Nikon TiE fluorescence microscope equipped with NIS-Elements Advanced Research imaging software (version 4.20). For all RNAscope mFISH experiments, positive cells were called by manually counting RNA spots for each gene. Cells were called positive for a gene if they contained ≥3 RNA spots for that gene. Lipofuscin autofluorescence was distinguished from RNA spot signal based on the larger size of lipofuscin granules and broad fluorescence spectrum of lipofuscin. The following probe combinations were applied to label cell types of interest: (1) LTK: LTK (NM_002344.5) and LAMP5 (NM_012261.3); (2) GLP2R: GLP2R (NM_004246.2) and CUX2 (NM_015267.3); (3) FREM3: RORB (NM_006914.3) and FREM3 (NM_001168235.2); (4) CARM1P1: RORB and CARTPT (NM_004291.3); (5) COL22A1: RORB and COL22A1 (NM_152888.3); (6) Adamts2: Cbr3 (NM_173047.3), Neurod1 (NM_010894.2) and Cdh13 (NM_019707.4); (7) Rrad: Nr4a3 (NM_015743.3), Cux1 (NM_009986.4) and Cdh13; (8) Agmat: Pou3f2 (NM_008899.2), Igfbp7 (NM_001159518.1) and Coch (NM_001198835). Experiments were repeated on at least $n = 2$ donors per probe combination for both mouse and human.

**Quantification of human and mouse soma size**
Images of NeuN$^+$ stained sections from human MTG (1 section per donor for 5 donors) and mouse VISp (1 section per mouse for 3 mice) (described above) were imported into ImageJ for processing. Regions of interest (ROIs) were drawn around cell bodies and exported as .roi files for downstream processing. In both species, L4 is defined as a band of densely packed, small granular cells, and the upper bound of this band (which includes overlying large pyramidal cells) is treated as the border between L3 and L4. The border between L1 and L2 is defined as the sharp boundary between the cell-sparse zone of L1 and the is a cell-dense zone of L2. In mouse, the border between L2 and L3 is indistinguishable and not defined. In human MTG, the boundary between L2 and L3 can be closely approximated as transition from densely packed small pyramidal cells to less densely packed larger pyramidal cells, which is largely consistent among donors.

Soma areas were defined as the number of pixels contained in each ROI, scaled by the number of pixels per μm. Cortical depth was defined for each cell as the position of that cell centroid relative to pia (absolute depth) or relative to the L1/2 and L3/4 boundaries (scaled depth) at that position in the tissue. The number of neurons per mm$^2$ of L2-3 neocortex (absolute density) is the number of neurons per image scaled by the area of the image where cell counts were assessed. For measuring surface density and cell area across L2-3 cortical depth, L2-3 was split into 20 evenly sized bins and the relevant measurements within each bin were calculated independently per section (one section per donor) and the average and standard deviation across sections were reported. The first and last bins are omitted from plots as they display boundary effects. Relative (scaled) neuron density scales to 1 for each donor and is defined as the fraction of total neuron count in each bin. In human, a nadir of scaled density was identified at −0.575, which we define as a quantitative boundary between superficial and deep L3 in this manuscript.

**Analysis of data from dissociated cells and nuclei**
Reference data used in this study include dissociated excitatory cells (mouse) or nuclei (human) collected from human MTG[3] and mouse VISp[26], and are all publicly accessible at the Allen Brain Map data portal (https://portal.brain-map.org/atlases-and-data/rnaseq). In human, cells from the five previously identified L2-3 glutamatergic types were retained, subsampling to match the laminar distribution of neurons included in the Patch-seq dataset as closely as possible, leaving a total of 2,948 neurons from *LTK*, *GLP2R*, *FREM3*, *CARM1P1* and *COL22A1* t-types. In mouse, all neurons from the three L2/3 glutamatergic t-types (*Adamts2*, *Rrad* and *Agmat*) were retained. Datasets were visualized as follows. First, the top 2,000 most binary genes by beta score[3] were selected. Beta score is defined as the squared differences in proportions of cells or nuclei in each cluster that expressed a gene above 1, normalized by the sum of absolute differences plus a small constant ($\varepsilon$) to avoid division by zero. Scores ranged from 0 to 1, and a perfectly binary marker had a score equal to 1. Second, the Seurat pipeline[39,40] (more details below) was used to scale the data, reduce the dimensionality using principal component analysis (PCA) (30 PCs). These PCs were then used to generate a UMAP[61]. Finally, data and metadata such as cluster, subcluster, layer and gene expression were then overlaid onto this UMAP space using different colored or shaded points.

Cluster heterogeneity is defined as average observed variance explained by the first PC compared with permuted data after accounting for differences in the number of cells per cell type. To get this, we (1) randomly selected 80 cells from each cell type, (2) identified the 80 most variable genes using the FindVariableFeatures Seurat function with selection.method="vst", (3) performed PCA after removing outlier cells, (4) calculated the percent of variance explained by the first PC, (5) repeated steps 1–4 for 100 sets of data where the expression levels for each gene are shuffled across the 80 cells to break gene correlations but retain other gene statistics, and (6) identified the average and standard deviation of PC1 for observed versus permuted data. Cluster discreteness is defined as the average number of differentially expressed genes between a given type and each of the remaining homologous t-types (*LTK*, *GLP2R* and *FREM3* t-types in human; *Rrad*, *Agmat* and *Adamts2* t-types in mouse). In this case pairwise differential expression is defined using the de_score function in the scrattch.hicat R library[26] after subsampling each cluster to 80 cells, and only the genes with higher expression in the relevant cluster are considered. The getMarkers function from the genesorteR R library (https://github.com/mahmoudibrahim/genesorteR)[42] was used to identify genes differentially expressed genes between deep *FREM3* (f73 subtype; collected from L3 or L4 dissection), *COL22A1* and *CARM1P1* neurons, using all default parameters except quant = 0.7. To validate the cell selection for deep L3 (since sublaminar dissection was not performed on the dissociated nuclei data), this analysis was repeated on Patch-seq neurons from these three types collected in deep L3 (scaled depth < −0.575). GO enrichment analysis was performed using ToppGene[62] with default settings, and Bonferroni-corrected *P*-values are reported unless stated otherwise.

**Dataset curation**
Patch-seq cells were included in this dataset if they met the following criteria. All neurons: (1) had high-quality transcriptomic data, measured as the normalized summed expression (NMS, adapted from the single-cell quality control measures in ref. [63]) of 'on'-type marker genes greater than 0.4; and (2) retained a soma through biocytin processing and imaging such that an accurate laminar association could be made. In addition, mouse neurons were: (1) located within VISp; (2) either tdTomato$^-$ or tdTomato$^+$ from a line known to label glutamatergic neurons (that is, tdTomato$^+$ neurons from known inhibitory mouse lines were excluded); (3) mapped to L2/3 IT VISp *Rrad*, L2/3 IT VISp *Agmat*, or L2/3 IT VISp *Adamts2* using Seurat mapping (as described below); and (4) mapped to L2/3 IT VISp *Rrad*, L2/3 IT VISp *Agmat*, L2/3 IT VISp *Adamts2*, or L4 IT VISp *Rspo1* in a separate Seurat mapping analysis where only reads located within gene introns are considered for both datasets. This final filter removes Patch-seq cells that jointly express markers for GABAergic and glutamatergic cells, probably representing L2/3 GABAergic neurons contaminated with adjacent glutamatergic cells. We do not find examples of such cells in human, possibly owing to a much smaller sampling of GABAergic cells than in the mouse.

**Identifying t-types**
Due to the differences in gene expression between Patch-seq and dissociated cells (see Extended Data Fig. 3b and a companion mouse study[41]), we used transcriptomes of dissociated human nuclei[3] or

cells[26] as reference datasets for human and mouse, respectively, and mapped Patch-seq transcriptomes to the reference data to identify their cell types. Prior to data transfer, we filtered genes potentially related to technical variables. X and Y chromosomes were excluded to avoid nuclei mapping based on sex. Many mitochondrial genes have expression correlated with RNA-seq data quality in dissociated nuclei data[3], so nuclear and mitochondrial genes downloaded from Human MitoCarta2.0[64] were also excluded. We also find that Patch-seq cells often have high expression of non-neuronal marker genes, so any genes most highly expressed in a non-neuronal cell type are excluded. Finally, any genes showing at least four-fold higher expression in dissociated nuclei versus Patch-seq cells in the included cell types (or vice versa) were excluded as potentially platform dependent. In total 23,129 of 50,281 genes (46%) remained in human and a comparable fraction for mouse. Variable genes for mapping were selected as described above for dissociated nuclei data visualization, by using the top 2,000 remaining genes by beta score as input into the procedure described below.

For both species, we mapped Patch-seq datasets to the relevant dissociated cells or nuclei reference using Seurat V3 (https://satijalab.org/seurat/)[39,40] following the tutorial for integration and label transfer with default parameters for all functions, except when they differed from those used in the tutorial, and replacing variable gene selection with the genes described above. More specifically, we first defined a low (30)-dimensional PCA space of the dissociated cells or nuclei dataset and then project this onto the Patch-seq dataset. We then found transfer anchors (cells that are mutual nearest neighbours between datasets) in this subspace. Each anchor is weighted on the basis of the consistency of anchors in its local neighbourhood, and these anchors were then used as input to guide label transfer (or batch correction), as described previously[65]. We then scaled the data, reduced the dimensionality using PCA, and visualized the results with UMAP[61]. This process is done using the FindTransferAnchors and TransferData R functions, which provide both the best mapping cell type and a confidence score. For mouse data, the three homologous types did not provide a heterogenous enough reference dataset, and therefore a larger set of glutamatergic and GABAergic cell types was used as reference. Cell-type assignments for most cells were robust to choice of reference dataset and to changes in parameter settings. Some cells with expression levels intermediate to two cell types changed calls between different runs; however, the cell-type-level results presented are robust to these small changes.

Multiple variations of three different mapping strategies were tested: (1) Seurat (as described in this Article), (2) a variation of the tree-mapping strategy previously described[41] for analysis of GABAergic cell types in mouse Patch-seq and (3) a correlation-based strategy comparing expression of marker genes in Patch-seq versus cluster medians of the scRNA-seq data. In all cases, 75–95% of cells mapped to the same cell type when comparing pairs of methods, consistent with similar analyses performed using dissociated FACS-sorted cells and therefore does not reflect a quality issue with Patch-seq data per se, but rather the fact that cell-type definitions are not totally discrete, and single cell measurements are not totally accurate (for example, owing to dropouts). As such, many cells show expression patterns that are not unambiguously mapped between highly similar transcriptomic cell types, although how much of this is biological and how much technical is difficult to assess. Despite the imperfect agreement of specific cells, the statistical results regarding morphology and electrophysiology for each cell type remain relatively unchanged regardless of the mapping method used.

Gene expression of Patch-seq cells was visualized by projection into the UMAP space calculated from dissociated nuclei using a combination of Seurat and the R implementation of the UMAP library (https://github.com/tkonopka/umap). More specifically, the Seurat data integration pipeline (functions FindIntegrationAnchors and IntegrateData) was used to calculated scaled data for both datasets and PCA was performed on this integrated space. The first 30 PCs from both datasets, as well

as the UMAP coordinates calculated for dissociated nuclei above were input into the UMAP pipeline and the 'predict' function was used to project the Patch-seq cells into UMAP coordinates. As above, data and metadata were then overlaid on these UMAP coordinates.

Since dissociated nuclei were not collected using sublaminar dissections, 'deep *FREM3*' neurons were defined as *FREM3* neurons dissected from L3 or L4 that were assigned to subtype f73 (Extended Data Fig. 1), which colocalizes with deep *FREM3* Patch-seq neurons in UMAP space (Figs. 1, 3). Furthermore, 77 of these 219 marker genes (including four genes shown in Fig. 4e) were also defined as marker genes by Patch-seq, where cortical depth was explicitly measured, suggesting the selection of deep *FREM3* neurons in dissociated nuclei was reasonable.

## Assessing transcript contamination

To quantify the effect of contamination and gene dropout in the Patch-seq dataset, we compared median gene expression levels of homologous t-types between platforms (Fig. 3a). Dissociated nuclei and Patch-seq cells from matched human t-types were highly correlated ($R = 0.85$, $P \approx 0$). Relatively few genes (177 genome-wide) showed enriched expression in dissociated nuclei relative to Patch-seq cells, suggesting that high quality transcriptomes collected in this dataset do not show the increased dropout rate reported in our previous study in mouse[41]. This is likely to be because we compared our human Patch-seq cells to a reference of dissociated nuclei, rather than the reference based on dissociated cells in mouse. By contrast, we identified 2,670 genes with at least fourfold enrichment in Patch-seq, including genes associated with extra-nuclear compartments such as the mitochondria ($P < 10^{-12}$) and ribosome ($P < 10^{-9}$), genes regulating cell death ($P < 10^{-18}$), RNA-binding genes ($P < 10^{-8}$) including immediate early genes, and markers for non-neuronal cells such as microglia ($P < 10^{-20}$). Some of the top genes in these categories include *COX3*, *FOS* and *IL1B*, which all show >100-fold enrichment in Patch-seq cells. These results indicated that Patch-seq cells probably contain some RNA collected from extranuclear compartments and from nearby contaminating cells (particularly microglia), and may show some activity-dependent transcription. However, these effects were minor compared to cell-type differences and we find overall high consistency and similar quality between Patch-seq cells and dissociated nuclei.

## Comparison of gene expression between species

Gene orthologues between mouse and human were pulled from the gene orthologues table on NIH (https://ftp.ncbi.nlm.nih.gov/gene/DATA/gene_orthologs.gz) on 22 November 2019. Only genes with unique orthologues between mouse and human were included in cross-species analyses.

## Electrophysiology feature analysis

Electrophysiological features were measured from responses elicited by short (3 ms) current pulses and long (1 s) current steps as previously described[9]. In brief, action potentials were detected by first identifying locations where the smoothed derivative of the membrane potential ($dV/dt$) exceeded 20 mV ms$^{-1}$, then refining on the basis of several criteria including threshold-to-peak voltage, time differences and absolute peak height. For each action potential, threshold, height, width (at half-height), fast after-hyperpolarization (AHP) and interspike trough were calculated (trough and AHP were measured relative to threshold), along with maximal upstroke and downstroke rates $dV/dt$ and the upstroke/downstroke ratio (that is, ratio of the peak upstroke to peak downstroke). Additional features from suprathreshold sweeps included the rheobase and slope of the firing rate versus current curve ($f$–$I$ slope); the first spike latency and initial firing rate (inverse of first inter-spike interval), measured at rheobase; and the mean firing rate and spike frequency adaptation ratio (mean ratio of consecutive inter-spike intervals), measured at ~50 pA above rheobase. Subthreshold features included the resting membrane potential (RMP), input

resistance and membrane time constant (tau) from response across or before hyperpolarizing long steps, and sag ratio from response at ~−100 pA. All feature calculation used the IPFX package (https://github.com/AllenInstitute/ipfx).

## Morphology feature analysis

Morphological features were calculated as previously described[9]. In brief, feature definitions were collected from prior studies[10,66]. Features were calculated using the version of neuron_morphology package (https://github.com/alleninstitute/neuron_morphology/tree/dev). Reconstructed neurons were aligned in the direction perpendicular to pia and white matter. Additional features, such as the laminar distribution of axon, were calculated from the aligned morphologies. Shrinkage correction was not performed (see above), features predominantly determined by differences in the z-dimension were not analysed to minimize technical artifacts due to z-compression of the slice after processing.

## Analysis of features by t-type and species

Combined datasets of electrophysiological and morphological features across homologous t-types from mouse and human were visualized by an analysis pipeline of data imputation and standardization, followed by projection to two dimensions using UMAP or SPCA (sklearn and umap python packages)[67,68]. Cells with more than 3 out of 18 electrophysiological features missing were dropped, the remaining missing features were imputed as the mean of 5 nearest neighbours (KNNImputer), and features were centred about the median and scaled by interquartile range (RobustScaler). The SPCA regularization parameter was adjusted to minimize non-zero features while preserving dataset structure. All features with coefficients over 0.05 were reported directly in the case of electrophysiology or summarized by feature categories for morphology.

For each feature, differentiation by t-type was assessed by running a one-way ANOVA for the feature by t-type, using the statsmodels package[69]. This analysis was repeated separately for the three mouse and human homologous t-types, as well as the three deep human t-types (with the subset of deep FREM3 cells only). Results were reported as fraction of variance explained ($\eta^2$ or $R^2$) and heteroscedasticity-robust $F$ test $P$-value (HC3), corrected for FDR (Benjamini–Hochberg procedure) across all features for each data modality. Post-hoc Mann–Whitney $U$ tests were run across pairs of t-types in each group (human and mouse homologous types and deep human types) for top-ranked features from ANOVA, and results FDR-corrected.

For classification of t-types, features were normalized using the standard scaler scalar in sklearn (StandardScalar), and the data were randomly assigned with stratification to training (70%) and testing sets (30%). The random forest classifier was trained using the sklearn package with 600 decision trees. The classification performance was estimated after averaging the results of the classifiers trained on 1,000 random data splits and compared against performance for data with shuffled t-type labels. Confusion matrices shown are for a single representative train–test split.

## Analysis of features by depth for FREM3 t-type

For each electrophysiology, morphology, and gene feature, the depth-related variability was assessed by a linear regression of the feature against relative L2-3 depth, using the statsmodels package[69]. Results were reported as fraction of variance explained ($R^2$), Pearson correlation $r$, and heteroscedasticity-robust $F$ test $P$-value (HC3), corrected for FDR (Benjamini–Hochberg procedure) across all features for each data modality. Owing to the large number of morphology and genes tested, results were summarized by calculating GO-term enrichment in ToppGene[62] for the set of depth-correlated genes (FDR < 0.05), followed by subselection of representative GO terms using REViGO[70]. Groups of features were ranked by the group's highest $R^2$, and the features with highest correlation shown for the top groups.

## Reporting summary

Further information on research design is available in the Nature Research Reporting Summary linked to this paper.

## Data availability

Transcriptomic, electrophysiological, and morphological data supporting the findings of this study are available at https://portal.brain-map.org/explore/classes/multimodal-characterization.

## Code availability

The custom electrophysiology data acquisition software (MIES) is available at https://github.com/alleninstitute/mies. The Vaa3D morphological reconstruction software, including the Mozak extension, is freely available at www.vaa3d.org and its code is available at https://github.com/Vaa3D. Code for reproducing most of the analyses presented in this work is available on GitHub https://github.com/AllenInstitute/patchseq_human_L23.

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

**Acknowledgements** We thank A. Wanner for providing reconstruction services through A. Szeto and R. Szeto, and Z. Popovic for facilitating the reconstruction work contributed by Mozak.science. We also thank the Mozak citizen scientists for their valuable contribution. The research was partially supported by several grant awards from institutes under the National Institutes of Health (NIH), including award U01MH114812 from National Institute of Mental Health, R01EY023173 from The National Eye Institute and U01MH105982 from the National Institute of Mental Health and Eunice Kennedy Shriver National Institute of Child Health & Human Development. The content is solely the responsibility of the authors and does not necessarily represent the official views of NIH and its subsidiary institutes. Work was supported by the Hungarian Academy of Sciences, the National Research, Development and Innovation Office of Hungary GINOP-2.3.2-15-2016-00018, the Ministry of Human Capacities of Hungary

20391-3/2018/FEKUSTRAT to G.T. Neuropathology support was provided in part by the Nancy and Buster Alvord Endowment to C.D.K. Work was supported by the European Union's Horizon 2020 Framework Programme for Research and Innovation under the specific grant agreement no. 945539 (Human Brain Project SGA3), ERANET programme iPS&BRAIN, and NWO Gravitation program BRAINSCAPES: A Roadmap from Neurogenetics to Neurobiology (NWO: 024.004.012). This work was funded by the Allen Institute for Brain Science. We dedicate this paper to the vision, encouragement, and long-term support of our founder, Paul G. Allen.

**Author contributions** I.S., H.D.M., G.T., H.Z., C.K. and E.S.L. conceptualized the project. J.T.T., N. Dee, A. Beller, T. Casper, P.C., R.D., N.H., C.H., L. Keene, C. Latimer, D. Marshall, E.M., J.N., A. Olah, J.-G.Y., P.B., C.C., P.C.d.W.H., R.G.E., M.F., R.P.G., S.I., C.D.K, A.L.K., J.G.O., A.P.P. and D.L.S. contributed to neurosurgical tissue acquisition. N. Dee, E.B., T. Casper, P.C., K.C., M.K., G.M., A. Ozsvar, J.S., H.T. and G.T. prepared brain slices. K. Baker, S.F., A.A.G., N.A.G., K.H., T.S.H., D.B.H., D.H., L. Kim, R.M., E.J.M., N.M., G.M., Lindsay Ng, G.O., A. Olah, A. Ozsvar, R.R., J.T., R.W. and G.T. performed electrophysiological experiments. D.B., A. Glandon, J.G., D. McMillan, T.P., C.R., M.T., A.T., K.W. and K.S. prepared scRNA-seq libraries. K. Bickley, J. Bomben, K. Brouner, T.E., A. Gary, H.G., M. Maxwell, M. McGraw, L.M., V.O., D.P., C.A.P. and A.R. processed slices for biocytin staining and immunohistochemistry. S.D.L., N. Dotson, R.E., M.G., M.H., K.N., P.R.N., V.O., L.P. and S.R. imaged immunostained and biocytin-stained slices and cells. C.D.K., C. Latimer and D. Marshall performed neuropathology scoring. S.-L.D., L.A., R.D., R.d.F., T.D., A.M.H., S.K., A.M., D.S. and G.W. reconstructed neurons and/or provided anatomical annotations. J. Berg, S.A.S., J.T.T., J.A.M., T. Chartrand, A. Buchin, A. Budzillo, S.-L.D., R.D.H., B.K., B.R.L., O.F., C.G., L. Keene, K.E.L., M. Mallory and Z.Y. performed analyses. T. Chartrand, A. Buchin, J. Berg, S.A.S., J.T.T., B.K., B.R.L., P.C., E.R.T. and A. Bernard contributed to method development studies. A. Budzillo, T.B., L.G., T.J. and D.R. generated tools for pipeline data generation. F.D., S.M., S.M.S. and L.E. provided programme management support. J. Berg, S.A.S., N. Dee, B.R.L., R.D., T.D., C.F., M. McGraw, P.R.N., L.P., G.J.M. and H.Z. organized and managed pipeline data generation. D.F., S.G., A.S., W.W. and Lydia Ng organized and managed pipeline data storage and processing. J. Berg, S.A.S., J.T.T., J.A.M., T. Chartrand, A. Buchin, T.E.B., A. Budzillo, R.D.H., B.K., B.R.L., N.W.G., M.-H.K., B.P.L., Z.Y., C.A., A.A., A. Bernard, P.R.H., A.R.J., G.J.M., J.W.P., R.Y., I.S., C.P.J.d.K., H.D.M., G.T., H.Z., C.K. and E.S.L. provided scientific direction. J. Berg, S.A.S., J.T.T., J.A.M., T. Chartrand, A. Buchin, A. Budzillo, S.-L.D., R.D.H. and E.S.L. prepared the figures. J. Berg, S.A.S., J.T.T., J.A.M., T. Chartrand, A. Buchin, C.A. and E.S.L. wrote the manuscript in consultation with all other authors.

**Competing interests** The authors declare no competing interests.

**Additional information**
**Correspondence and requests for materials** should be addressed to Ed S. Lein.

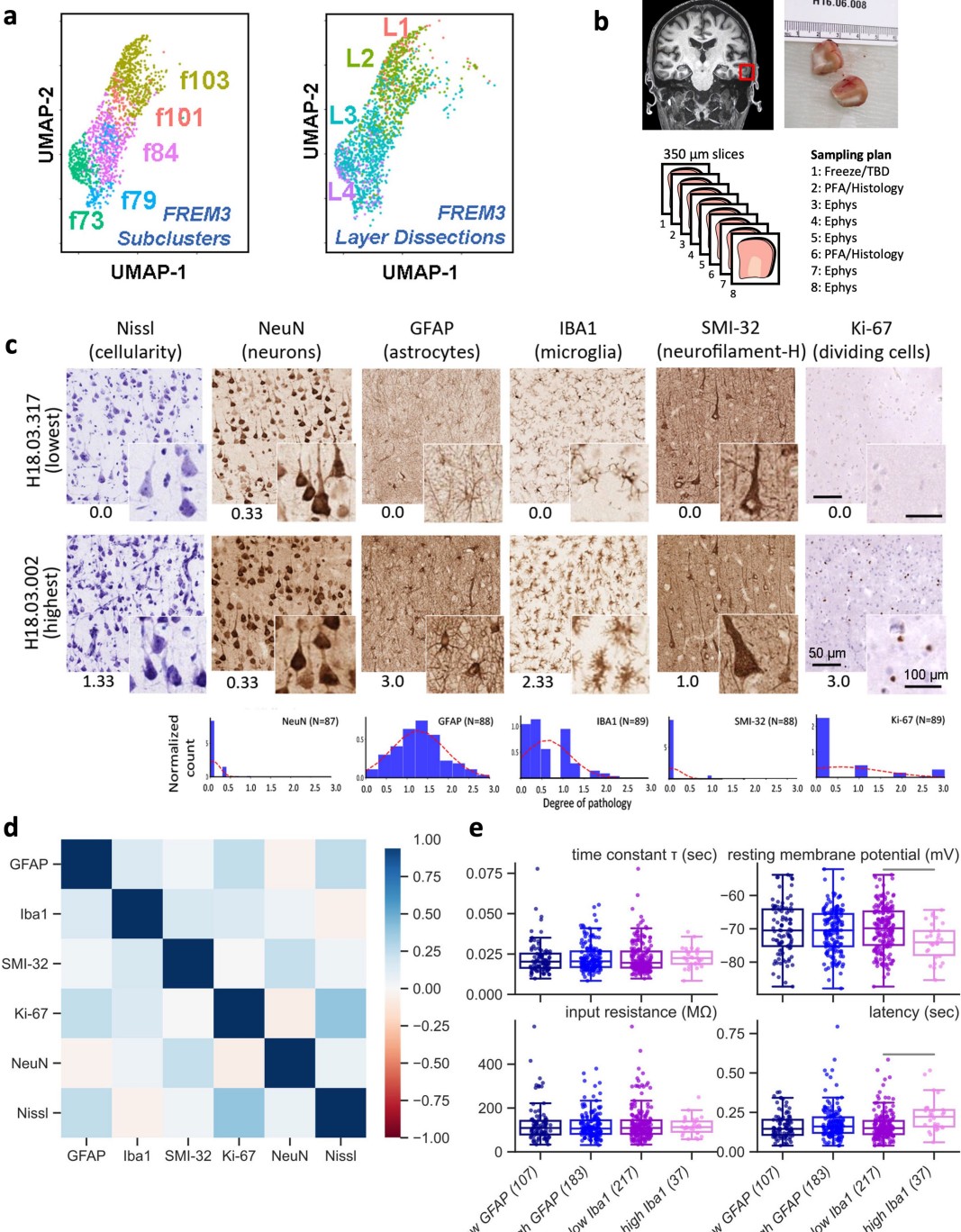

**Extended Data Fig. 1 | Human tissue acquisition and pathology analysis.**
**a**, UMAP of 2,948 dissociated human nuclei collected26 from five glutamatergic t-types in L2 and L3 of human MTG using the 2,000 most binary genes (repeated from Fig. 1d for clarity). FREM3 nuclei, color-coded by subtype assignment 26(middle) or dissected layer (right). **b**, Example resected tissue specimen from human middle temporal gyrus is processed into a series of 350 μm-thick slices according to a standardized sampling plan. **c**, Immunohistochemistry and imaging on human surgical specimens. Averages of scores from 0 [normal] to 3 [pathological]. Shown are images for donors with the lowest (top) and highest (middle) average marker score. Scores indicated below each image. Bottom, histograms of scores across all donors (N=number of cases). **d**, Pearson correlation coefficient between various tissue pathology scores: GFAP, IBA1, SMI-32, Ki-67, NeuN and Nissl. **e**, Boxplots of electrophysiology features with potential relationships to pathology. Cells are assigned to low or high pathology groups based on pathology scores <1 or ≥1 respectively. Bars indicate significant pairwise comparisons (p<0.05, FDR-corrected Mann-Whitney test), both of which are nonsignificant once cell depth is included as a factor (main text). Boxes show median (center) and quartiles (ends), whiskers show trimmed range bounded at 1.5×IQR beyond quartiles.

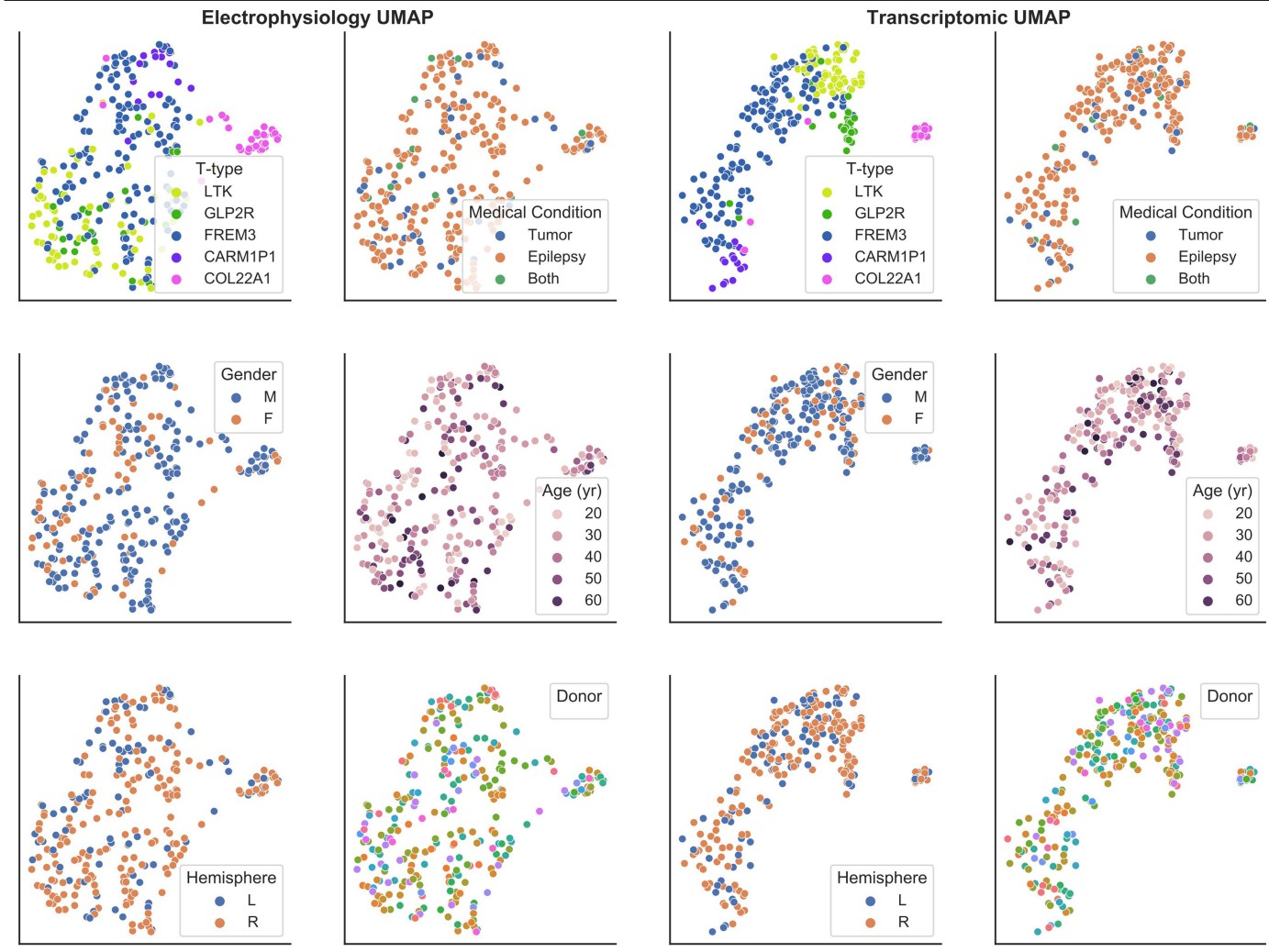

**Extended Data Fig. 2 | Relationships between patient metadata and features.** UMAP projection of electrophysiological features (left) and gene expression (right), with data points for each neuron colored by t- type (upper left) and by patient characteristics. In particular, cells split by medical condition (upper right) show a lack of correspondence between pathology, electrophysiology, and transcriptomic cell identity.

**a**

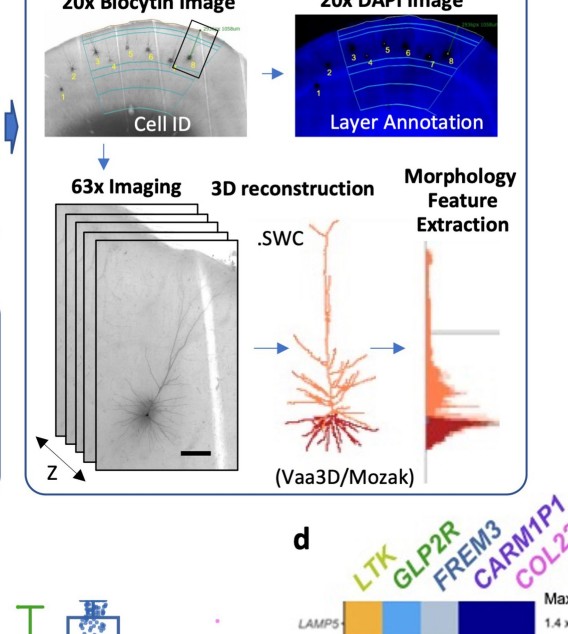

## 1. Patch-clamp recording and biocytin fill

**Standardized Stimulus Set with Integrated QC**

**Analysis pipeline: Intrinsic Physiology Feature Extractor (IPFX)**

1 s square 3 ms square

## 2. Transcriptomics from extracted nuclei

**Nucleus Extraction** **Reverse transcription** **cDNA Amp.**

RT A A A A / T T T T

(SMART-Seq v4)

Sequencing

## 3. Imaging and neuron reconstruction

**20x Biocytin Image** **20x DAPI Image**

Cell ID Layer Annotation

**63x Imaging** **3D reconstruction** **Morphology Feature Extraction**

.SWC

(Vaa3D/Mozak)

**b**, **c**, **d**

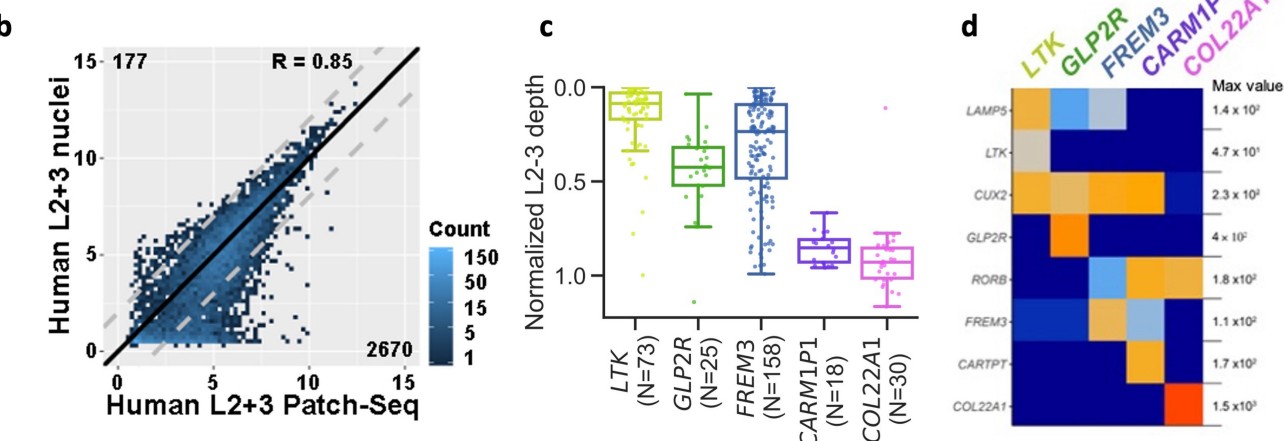

**Extended Data Fig. 3 | Human Patch-seq pipeline. a**, Workflow for patch clamp recording using standardized stimulus protocols and feature extraction code (1), followed by RNA-seq on extracted nucleated patches (2). Biocytin-filled neurons in slices are visualized with DAB as chromogen, imaged, and digitally reconstructed for morphological feature calculation and analysis (3). **b**, Density scatter plot showing the average expression of genes between dissociated nuclei and Patch-seq cells in human. Dashed lines indicate two-fold enrichment, with number of differentially expressed genes shown in the off-diagonal corners. p~0. **c**, Depth distribution of neurons in human and mouse supragranular layers normalized to depth within L2-3, grouped and colored by t-type. All pairwise comparisons are significant at FDR<0.05 (Mann-Whitney test). Boxes show median and quartiles, whiskers show trimmed range without outliers >1.5 IQR beyond quartiles. Individual neuron data points horizontally jittered for clarity. **d**, Marker gene expression values for each t-type, based on FACS data[3], shown for all five human t-types, normalized by gene.

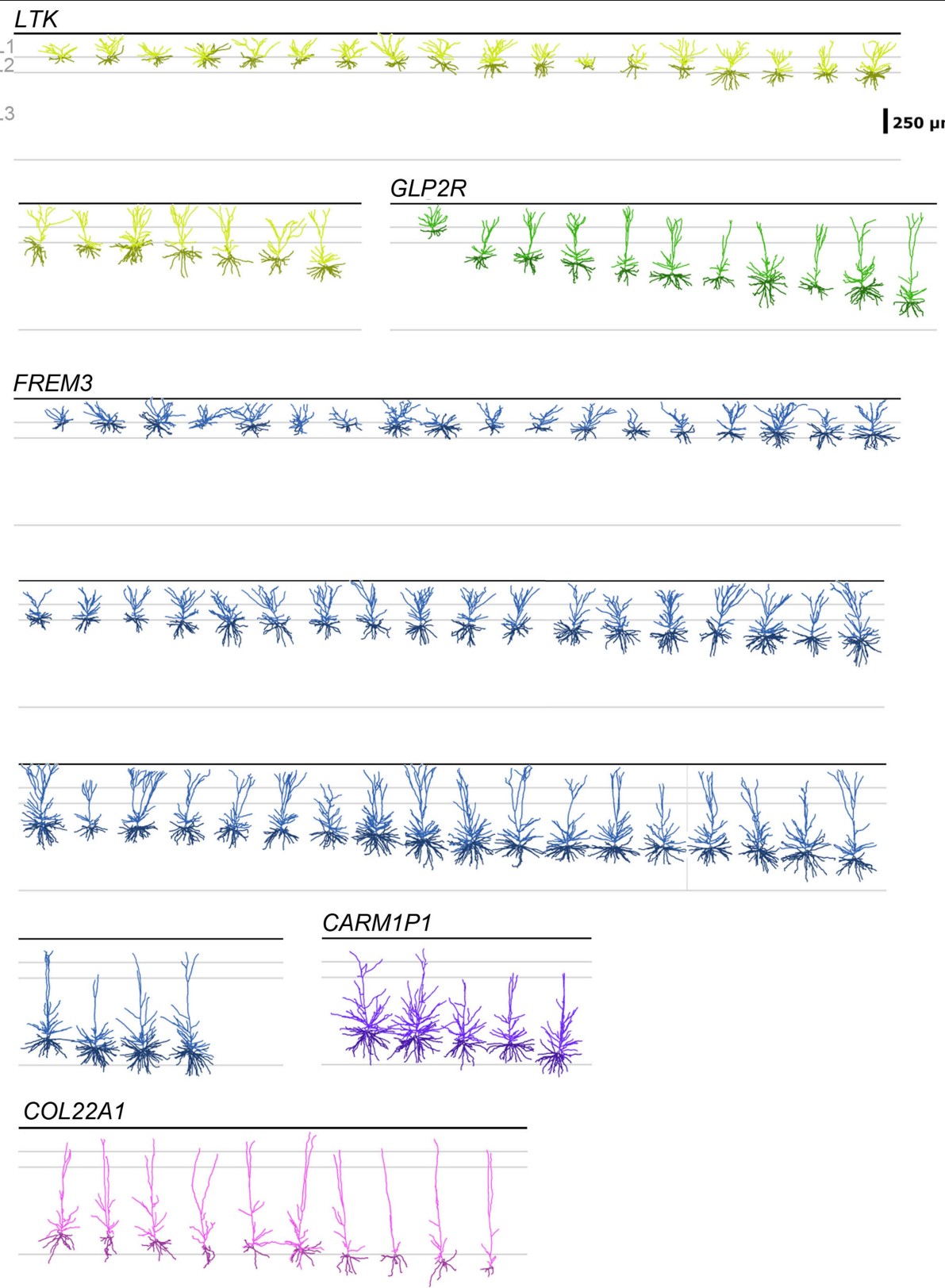

**Extended Data Fig. 4 | Human L2-3 excitatory neuron dendritic reconstructions.** All human L2-3 excitatory neuron dendritic reconstructions ordered by t-type and aligned by layer to an average cortical template. Apical dendrites are in darker colors, basal dendrites in lighter colors. The division between superficial and deep *FREM3* neurons is indicated by the gray vertical line.

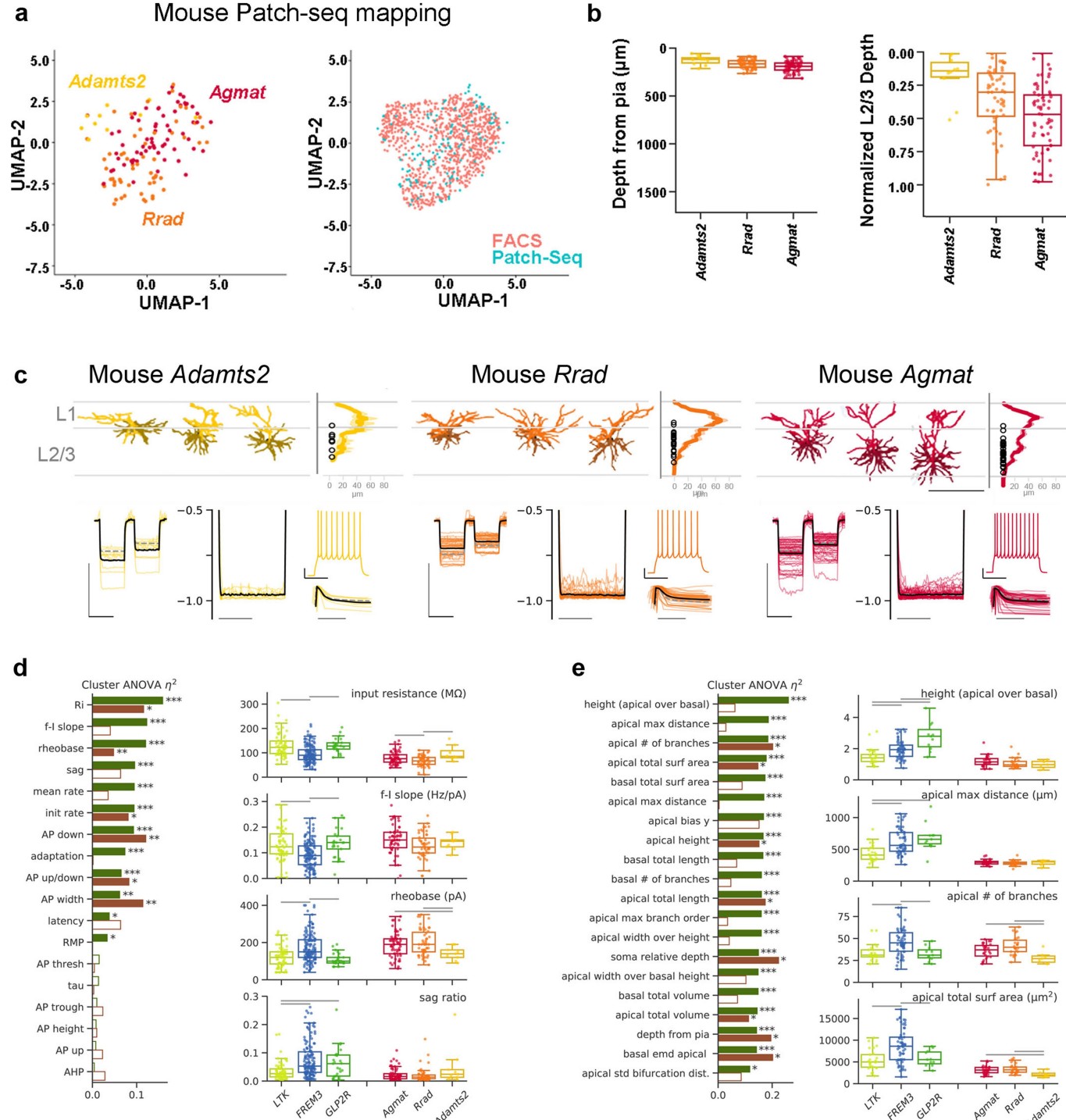

**Extended Data Fig. 5 |** See next page for caption.

**Extended Data Fig. 5 | Mouse VISp L2/3 excitatory neurons are less morphoelectrically discrete than their homologous human L2, L3 counterparts. a**, Joint UMAP of dissociated mouse cells from Fig. 1e and 133 glutamatergic Patch-seq neurons from supragranular cortex in VISp. Left plot shows cells color-coded by collection strategy. Right plot shows only Patch-seq neurons color-coded by mapped t-type. **b**, Depth distribution of neurons in mouse supragranular cortex, grouped and colored by t-type. Left plot shows depth from pia in μm. Right plot shows scaled depth within L2/3. Boxes show median and quartiles, whiskers show trimmed range without outliers >1.5 IQR beyond quartiles. Individual neuron data points horizontally jittered for clarity. **c**, Morphology and electrophysiology descriptions of the three L2/3 glutamatergic t-types in mouse visual cortex: *Adamts2*, *Rrad*, and *Agmat*. For each panel, colored lines are individual neurons, solid black line represents the mean of all neurons in that t-type, dashed gray line represents the global mean of the other 2 homologous t-types in that species. Left is an overlaid response to -70 and -30 pA current injections (scale bar = 10 mV, 1.0 s), center left are hyperpolarizing pulses normalized to their peak deflection to allow for a sag comparison, shown is the range -0.5 to -1.0 (scale bar = 0.5 s). Right is a representative suprathreshold spiking response (top, scale bar = 20 mV, 0.5 s), and the normalized instantaneous firing rates for a suprathreshold pulse, demonstrating the neuron's firing rate adaptation (bottom, scale bar = 0.5 s). Scale bar = 250 μm. Electrophysiological responses are shown for 9 *Adamst2*, 43 *Rrad* and 55 *Agmat* cells. **d**, **e**, Effect size (explained variance) for one-way ANOVA of each electrophysiology (**d**) and morphology (**e**) feature vs. t-type for human (green) and mouse (red). Stars indicate significance at FDR (False Discovery Rate) < (0.05, 0.01, 0.001). Box plots on right show data distribution by t-type for the four features with the largest effect size in human. Gray bars indicate significant pairwise comparisons (FDR<0.05, Mann-Whitney test). Boxes show median and quartiles, whiskers show trimmed range without outliers >1.5 IQR beyond quartiles. Individual neuron data points horizontally jittered for clarity.

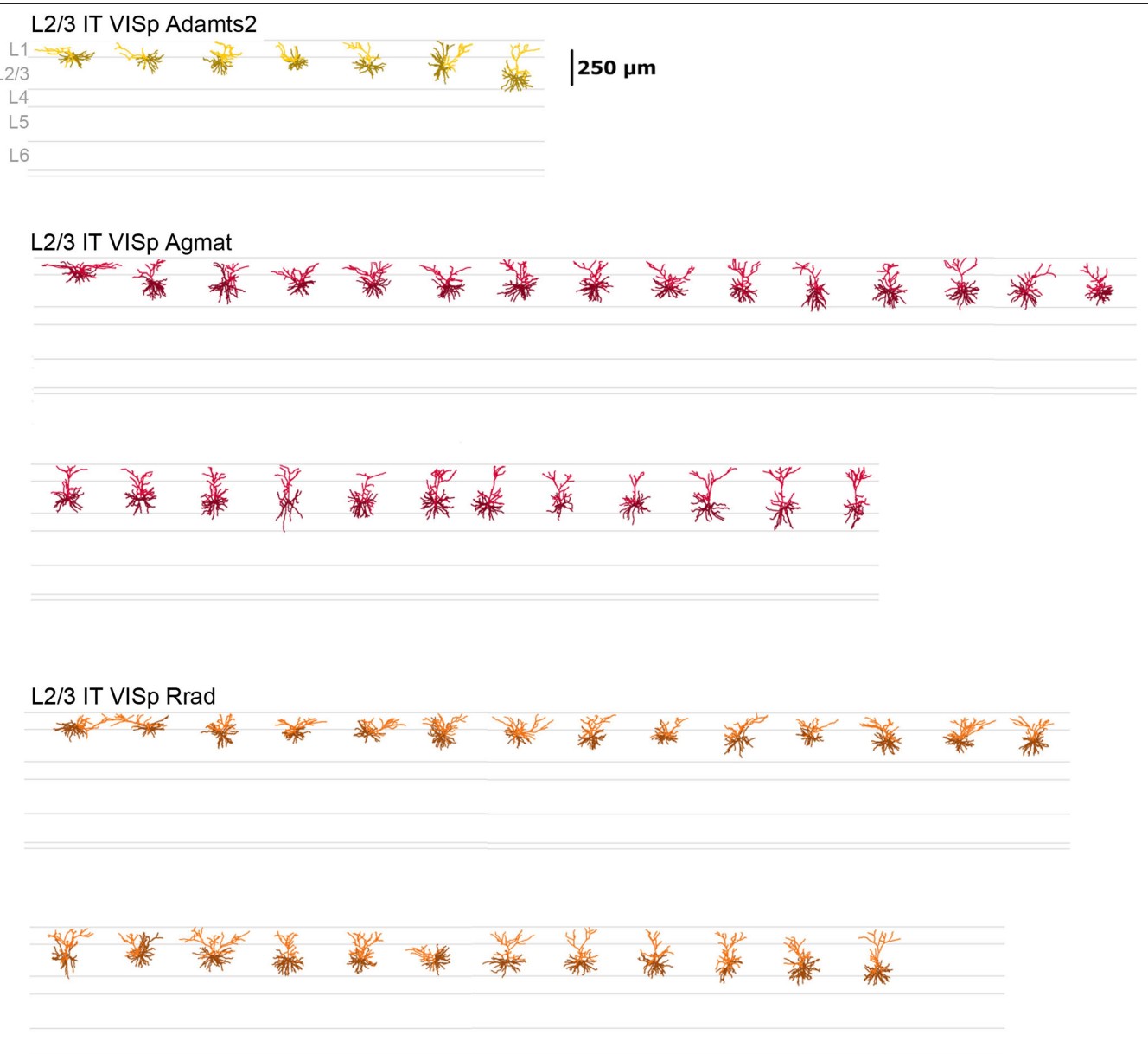

**Extended Data Fig. 6 | Mouse L2/3 excitatory neuron dendritic reconstructions.** All mouse L2/3 excitatory neuron dendritic reconstructions ordered by t-type and aligned by layer to an average cortical template. Apical dendrites are in darker colors, basal dendrites in lighter colors.

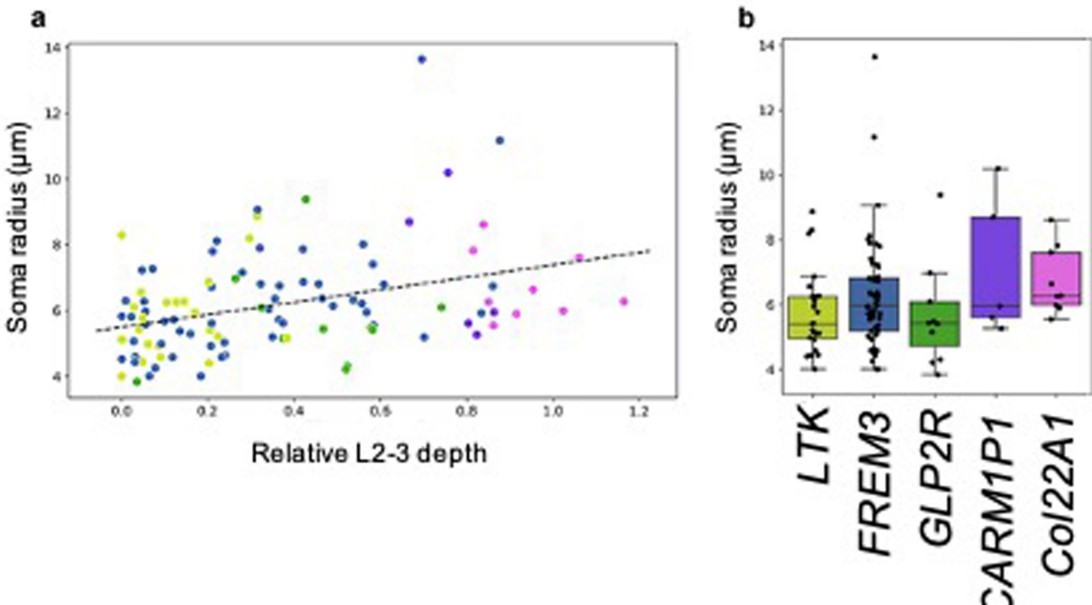

**Extended Data Fig. 7 | Somata radius by depth and t-type. a**, Soma radius vs. normalized L2-3 depth. Each soma is colored by t-type. **b**, Average soma radius by t-type for human and mouse.

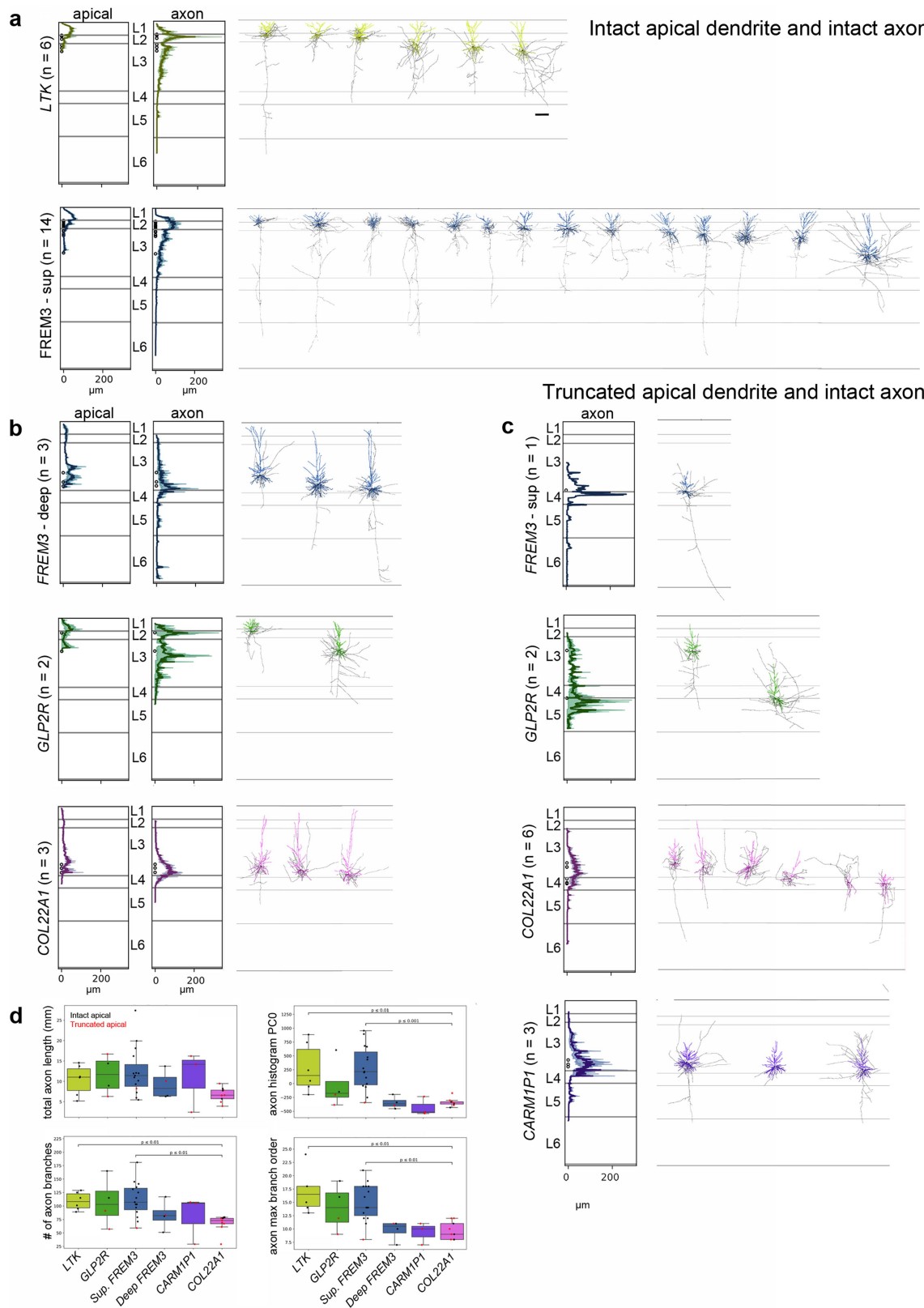

**Extended Data Fig. 8** | See next page for caption.

**Extended Data Fig. 8 | Axon distribution pattern by t-type. a**, Morphology descriptions of *LTK* and superficial *FREM3* neurons with intact apical dendrites and substantial local axon (top two rows). For each panel: Left, Histograms of the average apical dendrite and axon branch length (normalized to the maximum value for each t-type) by cortical depth and layer. Right, representative examples of morphological reconstructions from each t-type. Axons appear in gray. Morphology descriptions of deep *FREM3*, *GLP2R*, *CARM1P1* and *COL22A1* neurons with either intact apical dendrites (**b**) or truncated apical dendrites (**c**)

and substantial local axon. For each panel: Left, Histograms of the average apical dendrite and/or axon branch length (normalized to the maximum value for each t-type) by cortical depth and layer. Right, representative examples of morphological reconstructions from each t-type. Axons appear in gray. **d**, Box plots illustrating axon feature distribution by t-type. Total axon length, axon histogram Principal Component (PC) 0, number of axon branches and maximum branch order are shown. Brackets indicate significant pairwise comparisons (FDR<0.05, Mann-Whitney test). Scale bar = 250 μm.

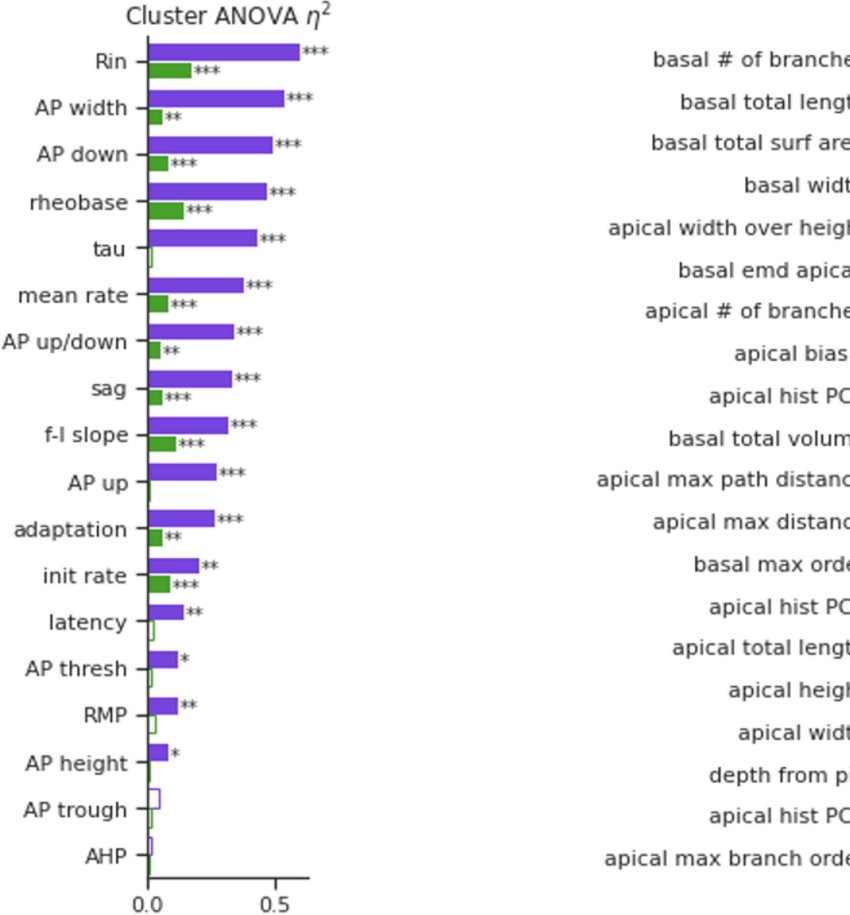

**Extended Data Fig. 9 | Deep human L2, L3 neuron types are more morphoelectrically distinct than superficial L2, L3 t-types.** Effect size (explained variance) for one-way ANOVA of each electrophysiology (left) and morphology (right) feature vs. t-type for human superficial L2-3 (*LTK*, *GLP2R*, Superficial *FREM3*, green) and deep L2-3 (Deep *FREM3*, *CARM1P1*, *COL22A1*, purple). Stars indicate significance at FDR (False Discovery Rate) < (0.05, 0.01, 0.001).

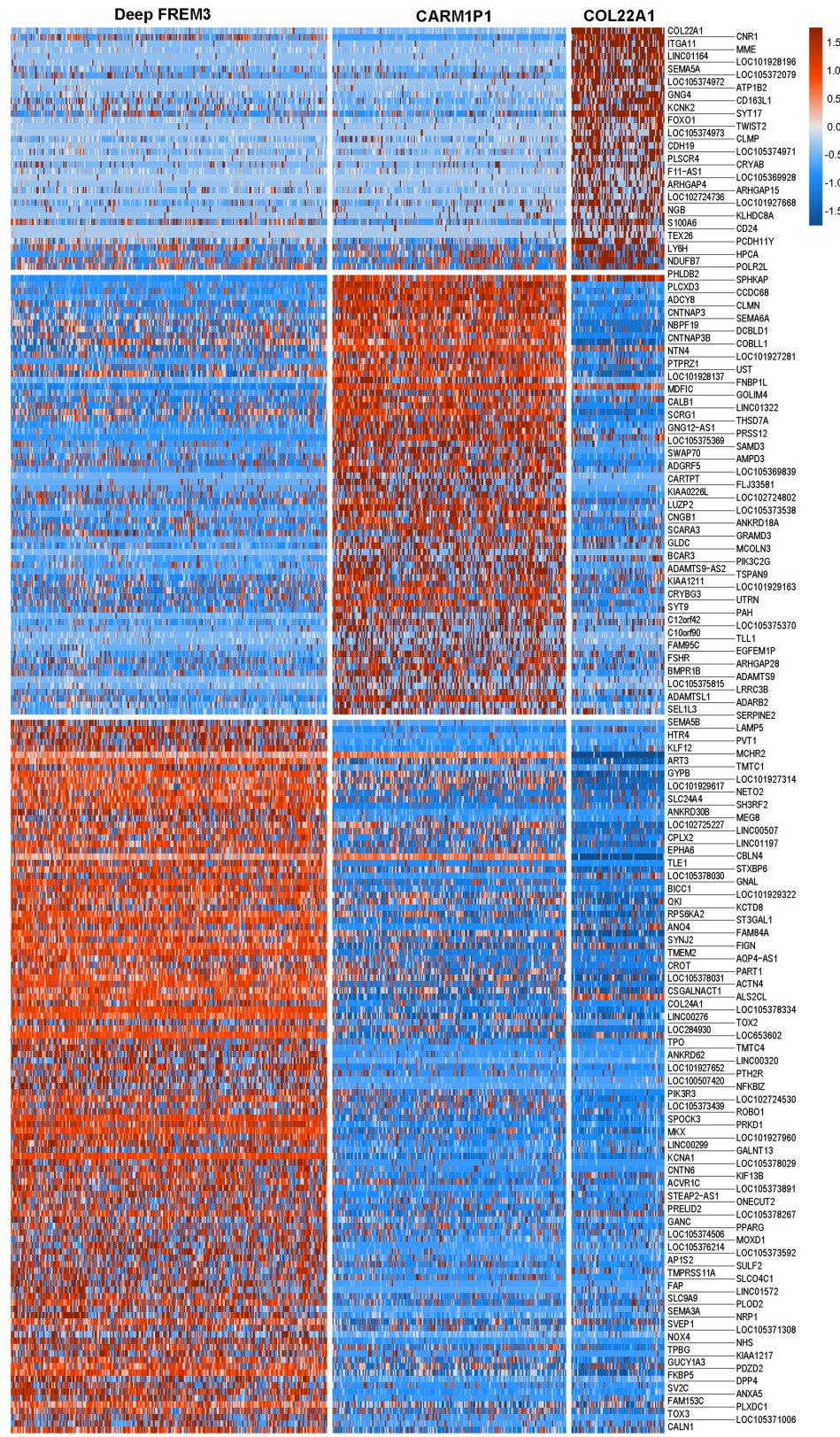

**Extended Data Fig. 10 | Differentially expressed genes between deep types.** differentially expressed genes selective for one or two of the *CARM1P1*, *COL22A1*, and deep *FREM3* t-types, selected using genesorteR. Heatmap and legend show Z score normalized expression values.

# nature research

# Reporting Summary

Nature Research wishes to improve the reproducibility of the work that we publish. This form provides structure for consistency and transparency in reporting. For further information on Nature Research policies, see Authors & Referees and the Editorial Policy Checklist.

## Statistics

For all statistical analyses, confirm that the following items are present in the figure legend, table legend, main text, or Methods section.

| n/a | Confirmed | |
|---|---|---|
| ☐ | ☒ | The exact sample size (*n*) for each experimental group/condition, given as a discrete number and unit of measurement |
| ☒ | ☐ | A statement on whether measurements were taken from distinct samples or whether the same sample was measured repeatedly |
| ☐ | ☒ | The statistical test(s) used AND whether they are one- or two-sided  *Only common tests should be described solely by name; describe more complex techniques in the Methods section.* |
| ☐ | ☒ | A description of all covariates tested |
| ☐ | ☒ | A description of any assumptions or corrections, such as tests of normality and adjustment for multiple comparisons |
| ☐ | ☒ | A full description of the statistical parameters including central tendency (e.g. means) or other basic estimates (e.g. regression coefficient) AND variation (e.g. standard deviation) or associated estimates of uncertainty (e.g. confidence intervals) |
| ☐ | ☒ | For null hypothesis testing, the test statistic (e.g. *F*, *t*, *r*) with confidence intervals, effect sizes, degrees of freedom and *P* value noted  *Give P values as exact values whenever suitable.* |
| ☒ | ☐ | For Bayesian analysis, information on the choice of priors and Markov chain Monte Carlo settings |
| ☒ | ☐ | For hierarchical and complex designs, identification of the appropriate level for tests and full reporting of outcomes |
| ☐ | ☒ | Estimates of effect sizes (e.g. Cohen's *d*, Pearson's *r*), indicating how they were calculated |

*Our web collection on statistics for biologists contains articles on many of the points above.*

## Software and code

Policy information about availability of computer code

| Data collection | Electrophysiology acquisition: Igor Pro 8 (Wavemetrics) with a custom acquisition module (https://github.com/AllenInstitute/MIES). Image acquisition: Zeiss Efficient navigation (ZEN) 2012 SP2 software module (Zeiss). |
|---|---|
| Data analysis | The Vaa3D version 3.2 morphological reconstruction software, including the Mozak extension, is freely available at www.vaa3d.org and its code is available at github.com/Vaa3D. The Python 3 code for electrophysiological and morphological feature analysis is available as part of open-source repositories (github.com/AllenInstitute/AllenSDK, github.com/alleninstitute/ipfx, github.com/alleninstitute/neuron_morphology). <br> Mapping and other transcriptomic analysis was performed using custom software written in R (3.6.1) which used several packages including cowplot (1.0.0), data.table (1.12.4), dplyr (0.8.3), feather (0.3.5), future (1.14.0), genesorteR (0.4.2), ggbeeswarm (0.6.0), ggplot2 (3.2.1), gplots (3.0.1.1), gridExtra (2.3), Matrix (1.2.17), matrixStats (0.55.0), mfishtools (0.0.1), pheatmap (1.0.12), scrattch.hicat (0.0.23), scrattch.vis (0.0.212), Seurat (3.1.1), umap (0.2.4.1), VENcelltypes (0.1.0), and WGCNA (1.68). Sequence alignment was performed using STAR v2.5.3. <br> Analysis of electrophysiology and morphology data was performed using custom software written in Python (3.7), using several packages including numpy (1.15), pandas (0.2.4), scipy (1.4), scikit-learn (0.22), seaborn (0.9), statsmodels (0.11), and umap-learn (0.3.8). Morphological features were calculated using the version of neuron_morphology package (https://github.com/alleninstitute/neuron_morphology/tree/dev). <br> Soma area and density was calculated in NeuN+ neurons using custom software written in R (3.6.1) which used several packages including dplyr (0.8.3), ggplot2 (3.2.1), msir (1.3.2), and RImageJROI (0.1.1). <br> All custom code will be made available at github.com/alleninstitute/patchseq_human_L23. <br> Analysis of gene-depth correlations was performed using ToppGene (https://toppgene.cchmc.org/enrichment.jsp) and REViGO (http://revigo.irb.hr/), along with custom Python (3.7) software using the goatools package (0.9.9). |

For manuscripts utilizing custom algorithms or software that are central to the research but not yet described in published literature, software must be made available to editors/reviewers. We strongly encourage code deposition in a community repository (e.g. GitHub). See the Nature Research guidelines for submitting code & software for further information.

## Data

Policy information about availability of data

All manuscripts must include a data availability statement. This statement should provide the following information, where applicable:

- Accession codes, unique identifiers, or web links for publicly available datasets
- A list of figures that have associated raw data
- A description of any restrictions on data availability

Transcriptomic, electrophysiological, and morphological data supporting the findings of this study are available at https://portal.brain-map.org/explore/classes/multimodal-characterization.

# Field-specific reporting

Please select the one below that is the best fit for your research. If you are not sure, read the appropriate sections before making your selection.

☒ Life sciences ☐ Behavioural & social sciences ☐ Ecological, evolutionary & environmental sciences

For a reference copy of the document with all sections, see nature.com/documents/nr-reporting-summary-flat.pdf

# Life sciences study design

All studies must disclose on these points even when the disclosure is negative.

| | |
|---|---|
| Sample size | No statistical methods were used to predetermine sample sizes, but the sample sizes here are similar to those reported in previous publications.<br>The size of the final data set was determined based on reaching a number of cells per transcriptomic type on which meaningful statistics could be applied to determine phenotypic differences. For the most rare type we encountered, CARM1P1, this worked out to 5 neurons with intact dendrite morphologies. |
| Data exclusions | All cells included in this study were used in the transcriptomics analysis and passed the exclusion criteria laid out in the "Dataset curation" Methods section. Addition data exclusions were imposed for electrophysiology and morphology. To be included in electrophysiology analysis, each cell was required to pass a set of pre-established quality control metrics, including a > 1 GΩ seal recorded prior to break-in and the initial access resistance < 20 MΩ and < 15% of the input resistance. For an individual sweep to be included: (1) the bridge balance was < 20 MΩ and < 15% of the Rinput, (2) bias (leak) current 0 +/- 100 pA, (3) root mean square (RMS) noise measurements in a short window (1.5 ms, to gauge high frequency noise) and longer window (500 ms, to measure patch instability) < 0.07 mV and 0.5 mV, respectively and (4) the difference in the voltage at the end of the data sweep (measured over 500 ms of rest) and the voltage measured immediately prior to the stimulus onset < 1 mV. (For some human cells, after inspection of traces by eye, exceptions were made to cell-level QC and the sweep-level constraint on change in voltage relaxed from 1 to 2 mV.)<br>Cells were excluded from morphological reconstruction due to pre-established quality control metrics (poor fill, extensive truncation or breakage of axon or dendrites). |
| Replication | A total of 385 human Patch-seq cells from 82 individuals were used in the analysis, including cells from multiple types of surgeries and from multiple cortical areas. Data acquired from multiple mice from multiple litters per transgenic line surveyed. Extensive acquisition metadata as well as detailed white papers are reported as part of the Allen Cell Types Database (celltypes.brain-map.org); these additional details are intended to aid other laboratories if they seek to replicate the results presented in this study.<br>Each cell is recorded as an independent experiment and tissue was collected from 90 independent donors. The fact that so cells from so many donots map so reliably (see Extended Data Figure 2) supports the replicability of this study. |
| Randomization | Randomization is not relevant to this study as there was a single condition for all acquired data. |
| Blinding | Blinding is not relevant to this study as there was a single condition for all acquired data. |

# Reporting for specific materials, systems and methods

We require information from authors about some types of materials, experimental systems and methods used in many studies. Here, indicate whether each material, system or method listed is relevant to your study. If you are not sure if a list item applies to your research, read the appropriate section before selecting a response.

## Materials & experimental systems

| n/a | Involved in the study |
|---|---|
| ☐ | ☒ Antibodies |
| ☒ | ☐ Eukaryotic cell lines |
| ☒ | ☐ Palaeontology |
| ☐ | ☒ Animals and other organisms |
| ☐ | ☒ Human research participants |
| ☒ | ☐ Clinical data |

## Methods

| n/a | Involved in the study |
|---|---|
| ☒ | ☐ ChIP-seq |
| ☒ | ☐ Flow cytometry |
| ☒ | ☐ MRI-based neuroimaging |

## Antibodies

| Antibodies used | Primary mouse anti-Neurofilament H (SMI-32, Biolegend, 801701), Neu-N (Millipore #MAB377, 1:2000); SMI-32 (Biolegend #801704, 1:2000); GFAP (Millipore #MAB360, 1:1500); Parvalbumin (Swant #PV235, 1:2000); Iba-1 (Wako #019-19741, 1:1000); Ki67 (Dako #M724001-2, 1:200), goat anti-mouse IgG (H+L) Alexa Fluor conjugates (594 or 647, ThermoFisher Scientific A-11005 or 21235). |
|---|---|
| Validation | Anti-Neurofilament H, Neu-N; SMI-32; GFAP; Parvalbumin; and Iba-1 were verified via internal controls in every brain tissue slice - morphologies of expected cells to be stained were verified for each application. Ki67 stains rapidly dividing cells and is not expected to label healthy brain tissue, so a positive control slice was run using tonsil tissue for each application. |

## Animals and other organisms

Policy information about studies involving animals; ARRIVE guidelines recommended for reporting animal research

| Laboratory animals | Mice (male and female) between the ages of P45-P70 were maintained on the C57BL/6J background, and newly received or generated transgenic lines were backcrossed to C57BL/6J. |
|---|---|
| Wild animals | The study did not involve wild animals. |
| Field-collected samples | The study did not involve field-collected samples. |
| Ethics oversight | All procedures were carried out in accordance with Institutional Animal Care and Use Committee at the Allen Institute for Brain |

Note that full information on the approval of the study protocol must also be provided in the manuscript.

## Human research participants

Policy information about studies involving human research participants

| Population characteristics | Humans (male and female) between the ages of 18-85 were that were undergoing surgery for epilepsy (73%), tumor (21%), both epilepsy and tumor (3%), or other reasons (2%) are included in this study. Surgeries were performed on both hemispheres (left=33%; right=67%) and in all four cortical hemispheres, but primarily temporal cortex (85%). Most donors were of unknown (71%) or unspecified (5%) ethnicity, with the following breakdown for individuals of specified ethnicity: 90% Caucasian, 5% non-Hispanic or Latino, 3% Alaskan Native, and 1% African-American. Genetic information is not known. |
|---|---|
| Recruitment | Surgical specimens were obtained from local hospitals (Harborview Medical Center, Swedish Medical Center and University of Washington Medical Center) in collaboration with local neurosurgeons. Although bias could be present if there are biases in characteristics of individuals undergoing surgery for epilepsy or tumor, or of the subset of individuals undergoing surgery who choose to donor brain tissue, in this study we did not identify any relationship between cell properties and age, gender, neuropathology, or reason for surgery. |
| Ethics oversight | All patients provided informed consent and experimental procedures were approved by Harborview Medical Center, Swedish Medical Center and University of Washington Medical Center institute review boards before commencing the study. |

Note that full information on the approval of the study protocol must also be provided in the manuscript.

