## [Peer Review File · Nature]

Manuscript Title: Human neocortical expansion involves glutamatergic neuron diversification

Reviewer Comments & Author Rebuttals

Reviewer Reports on the Initial Version:

Referees' comments:

Referee #1:

This is a well-organized and technically strong manuscript as expected from the Allen Institute, which continues to produce superb resources and analyses. The paper intends to describe the ways in which the superficial layers of the human cortex have expanded during brain evolution compared with the mouse. While the concept of this paper affords an opportunity to do just that – an unprecedented cross species comparison with single-cell and PatchSeq approaches that provide the ability to match transcriptome, morphology and function in same cells in human tissue from surgery resections, the evolutionary comparison is not convincing and is not integrated with other far more convincing published demonstrations of primate and human brain evolution. Further, the authors do not efficiently compare and establish their work within the current knowledge of the field. Even with these large weaknesses, one does have to appreciate that this work addresses another important question – not necessarily anything to do with evolution, but the correspondence between transcriptional diversity and functional diversity at the physiological level as mentioned above should probably be the focus of the paper – how to define a cell type? Still this requires a far more earnest comparison to multiple mouse regions and other well accepted human single-cell cortical datasets. While it is understood that this is not an easy feat, the field needs a better way to ground the expansion of transcriptionally-defined cells onto regional, anatomical and functional features of neurons into commonly accepted types.

This paper extends previous work on single cell analysis, starting by concentrating on molecular comparisons between human and mouse to identify evolutionarily important differences. This comparison relies on more detailed analysis of data that has been published in two previous manuscripts that they reanalyze in more detail here. They use data from visual cortex in mouse, which is likely a sample of convenience, as it is not necessarily the most evolved or related to human/primate cognitive traits, and does not match with the human region profiled. Essentially the comparison is between a primary sensory region in one species with an association area in another. In fact, visual cortex is probably the most distinct cortical region in primates and one wonders what a direct comparison of human visual cortex with MTG would show?

That the mouse shows less regional difference than human in cortex does not remediate this issue as claimed in the introduction. This is quite a thin argument to support comparing vastly different brain regions to understand how the superficial layer of the human cortex evolved. While that statement is true for the magnitude of differential gene expression in bulk RNAseq experiments, where subtle changes in gene expression underlying cellular subtypes are not easily discovered, the simple amount of differential gene expression is not necessarily informative of cellular diversity. Any conclusions based on these comparisons only confuses the already rapidly growing complex landscape of transcriptionally defined cells. Because of these regional differences, the comparison between human and mouse is fraught from the beginning, and only slightly mitigated by comparisons between glut neurons that are transcriptionally matching. Lastly, the various sizes and thickness differences between species have been generally remarked on before and there is a literature on the size of human and primate cells and their extensive dendritic branching, etc. There is no question that the lamination of mouse cortex and human are markedly different, but

so is human visual cortex from MTG.

The work here also derives narrowly from theories and approaches seemingly based primarily on the authors internal work and lacks scholarly comparison to the field or historically what has been found. The authors downplay the absolutely enormous amount of prior work defining the diversity of glutamatergic cell types across different areas of the mouse cortex – specialized specifically for the mature circuit they function within such as intrinsic membrane properties, morphology, and transcriptomics. The authors downplay the diversity that is known in the across the mouse cortex and within excitatory neuron populations. Prior to the explosion of single-cell datasets, decades of work have gone into carefully characterizing cell types in the mouse model by basic criteria of location, intrinsic membrane properties and basic transcription factor cascades or markers selective for those cells (Functional and anatomical correlates: Fishell, Macklis, Arlotta, Marin, Rubenstein, McBain, Barres, Stevens, Krigstein, etc). Authors do not ground their discovered cell types in the vast literature that which define cells based on these multi-modal features. This is a HUGE loss given they have a potentially powerful and unique dataset composed of nuclear transcriptome sequencing, morphology and intrinsic membrane properties within human cells within the superficial layers cortex.

This dataset offers them a very special understanding of neurons at a level mostly unprecedented in the human, which is what I view as the main strength of the manuscript – the extensive anatomical and physiological characterization. Fundamentally the major contribution of the work is In the several t-type to other phenotype comparisons made – other related strengths of this work include: a) very thorough characterization of the cells they were able to profile through Patch-Seq and single-cell RNAseq in brain slices from human brain resections. b) development of an efficient scoring method for quantifying the pathology of tissue during analysis to signal healthy state of cells (as tissue comes from surgeries to remove tumors, epileptic centers, etc). c) Interrogating almost 300 glutamatergic neurons within the superficial layers of the human brain (from parietal, frontal, and temporal cortex) for three simultaneous modalities in each cell: nuclei- transcriptome, morphology, and functional membrane properties. This last point marks the key strength of this paper, which perhaps the entire paper should be refocused around: showing this correspondence in humans and relationship between molecular, anatomical/circuit placement and physiological features. The mouse-human part is just not remarkable – For example, a major finding shown in figure 4 is that human L2 and L3 t-types show “greater morphological and electrophysiological differentiation than homologue mouse t-types.” This is interesting, but not unexpected. Again, these comparisons need to be more firmly placed within current knowledge. This could be addressed computationally with existing datasets, but far more substantial comparisons are required for a true understanding of how diversification of the supragranular cortex evolved from mice to human.

On a more minor, but still important level, the authors focus on supragranular neurons and cite several findings that they claim show the importance of these layers in human evolution, including their vulnerability in Alzheimer’s disease. This putative relationship to Alzheimer’s Disease is somewhat of a non-sequitur, because it has little to do directly with human brain evolution – are the authors saying that this vulnerability is human specific, and that these pyramidal neurons are not affected in mice? -- because that has not been demonstrated. From this perspective, the Von Economo neuron would be more important to look at, which they do not. Further, the expansion of the cerebral cortex involves increased cell division and the expansion of the outer subventricular zone in primates and humans. There are specific regions of the genomes (regulatory) and genes (targets) that have been associated with the expansion of these cell classes in several publications and none of these are cited or discussed. How does the diversity of the cell types identified relate to identified sequence differences in regulatory regions or the genes that they regulate in the human genome? Again, the focus on these layers is well deserved based on their importance in human brain function and evolution, but some of the rationale and the actual human mouse comparison does not work well, and is not convincing.

Referee #2:

The manuscript "Human cortical expansion involves diversification and specialization of supragranular intratelencephalic-projecting neurons" by Berg...Lein contains many original and impressive results.

They include at least:

a) Stereotypy of neuron types, assessed by transcriptomic, morphological and electrophysiological combined approaches across several human neurosurgical specimens that are heterogeneous for many parameters such as age, gender, ethnicity, disease's condition and severity.

b) Demonstration that transcriptomes of neurons in human supragranular cortex are well-correlated with their morphological and physiological features, as well as cortical depth.

c) Identification of five novel neuron t-(transcriptional) neuron types in the human supragranular cortex. Specifically, LTK t-type that contains small neurons and is largely restricted to L2; GLP2R t-type that is found in L2 and superficial L3 with few dendrites and minimal branching in L2; FREM3 t-type that is found in L2 and L3, and shows continuous transcriptomic, morphological and physiological variation as function of cortical depth; CARM1P1 and COL22A1 t-types that are detected exclusively in deep L3, they have wide, highly branched apical dendrites and tall, very sparse apical dendrites, respectively.

d) Observation that deeper L3 FREM3 and CARM1P1 neurons are labelled by an SMI-32 antibody for subunits of neurofilament proteins. Notably this SMI-32 immunolabeling is lacking in Alzheimer's disease (AD), suggesting that AD progression could target specific cortical neuron types in humans (FREM and CARM1P1).

Overall, the results contained in this manuscript, represent a significant advancement in the understanding of human cortical glutamatergic neuron types. The novel information submitted, when it will be published, is likely to have a strong influence for future research in the field. As a whole, the data submitted also provide a remarkable demonstration that transcriptomic is a powerful approach that can be successfully used in combination with structural and functional approaches. The manuscript also provides an important, unprecedented and comprehensive quantitative analysis of neuronal properties in relation to cellular markers of pathology in the cortical samples taken from neurological patients.

I have the following constructive criticism.

1) A comparison is made between neurons of the human middle temporal gyrus (MTG) and mouse primary visual cortex (VISp). However, there is strong specialization of principal neurons according to cortical functional areas, which was also shown by one of the studies showing single cell data in the mouse (Tasic et al. 2018), where subtypes of principal neurons were almost exclusive either for lateral motor or primary visual cortex. Therefore, the choice of comparing L2/3 neuron types in the mouse visual cortex versus human MTG is sub-optimal. To compare homologous areas in the human and mouse cortex would improve this situation and is highly recommended for a new edition of the manuscript.

2) Electrophysiological characterization relies on several passive and active parameters that have been carefully collected and analyzed. However, spiking in response to squared depolarizing current pulses lasting several hundred milliseconds represent poorly defined physiological stimuli. They may occur "in real life" for some cells during cortical up-states and desynchronized brain states, but otherwise they are a quite artificial way (although widely used since the 50s - Granit's intracellular recordings of spinal motoneurons) to assess neuron's excitability. The manuscript

should acknowledge the relatively poor physiological meaning of this protocol.

The study would greatly benefit from an analysis of the synaptic inputs of the transcriptomically-defined neuron types. Due to the differential laminar distribution, shape and complexity of their dendrites, do these neurons receive similar amount of excitation and/or inhibition? What are the plastic properties of excitation and inhibition impinging on these neuron types? I am well aware that a single manuscript cannot address all the relevant questions. However, I do think that to submit at least some of synaptic data here, would make this manuscript more meaningful, stronger and interesting. Also, detailed information on dendritic patterns for each neuron type is given, but no attempts are made to discuss what inputs (cortical, sub-cortical etc) are likely to connect to the various neuron types. A discussion on this should be added.

3) It is a bit surprising that the manuscript does not provide any information on the axons of the morphologically reconstructed neurons. I understand that the axon is cut by the slicing procedure, yet some local axon collaterals should be detected within the slices. To include information on axon's aspect, branches, layers occupied and possibly targets etc, would represent a significant improvement. Some discussion on the possible targets of the newly-defined neuron types should be also added.

4) Data shown in Fig 2f. What could be the meaning of the more hyperpolarized membrane potential in the high IBA1 group (more pathological) compared to the other groups (less pathological, normal)?

5) The introduction could be improved. It mostly repeats the main results of the paper. Instead, it should provide more information on the state of the art of the field, background information and explain the rationale behind the methodologies used.

6) While the number of human cells for Patch-seq is rather high, the number of mouse cells studied is rather low, e.g. in Fig. 3c right panel <10 cells align to Adamts2 cluster. The Authors should increase the number of mouse Patch-seq neurons to provide a more robust comparison with human data.

7) Figure 1 data. Based on UMAP plot and panel e, FREM3 neurons might be further separated in t-subtypes. Did the Authors try to find markers that potentially separate FREM3 neurons in sub-clusters? Could be that 5 subtypes in MTG is an underestimation due to the relatively small number of cells analyzed?

8) Figure 3, Extended Figure 2. Representation of neurons in Patch-seq data should be shown per sample. While it is certainly difficult to label that many human samples at UMAP plots, such information could be provided as a table, where for each human sample the number of neurons for Patch-seq is given for each subtype. In order to be more confident that each subtype was found in many human samples, the number of human samples where a subtype was found should be indicated.

9) Figure 4, Extended Figure 7. Integration of mouse and human data is obviously a difficult task and from the results and methods submitted in this manuscript it is not completely clear how much effort has been spent for a robust integration of inter-species data. Did the Authors try to change data integration parameters by reducing some differences between mouse and human cells allowing for better inter-species alignment? The Authors used Seurat for data integration, however there are a number of other tools that performed well in comparative studies, did the Authors try some other tools?

10) Extended Figure 2, age panel. It is surprising to see lack of differences between neurons from 20 and 80-year-old brains. The Authors should comment this result in the text, and provide an interpretation for the lack of aged-related differences in the parameters studied.

11) In the discussion, the manuscript should attempt to better correlate the novel glutamatergic neuron types proposed by using transcriptomics, functional and anatomical analyses with the already known glutamatergic neurons of the supragranular cortex found in humans or other species, such as primates, cats or rodents.

Author Rebuttals to Initial Comments:

We thank the referees for their insightful and constructive comments, and general appreciation for the significance of the work for understanding human cortical cell types.

The single biggest issue raised by referees with the study as originally written was the comparison between human middle temporal gyrus (MTG) and mouse primary visual cortex. We fully appreciate the caveats that cross-areal introduces into a comparison between species. But, because of clinical limitations in access to human surgical samples, the only area where it is possible today to perform systematic *in vitro* studies consistently in human cortex is the middle temporal gyrus (MTG). We acknowledge that it is indeed challenging to compare directly to mouse neocortex given the lack of a truly homologous region. Nevertheless, the majority of our results carry across neocortical areas, and show common transcriptomic profiles that we and others are using as the foundation of our Patch-seq approach. We therefore strongly argue that the comparative framing of the work be preserved, as it illuminates biological findings in human compared to the dominant model organism in neuroscience research which go beyond the differences between cortical areas. To accommodate the referees' perspective on where the novelty of the study lies, in this new version, we have reframed the manuscript to establish that human transcriptomic cell types and axes of variation are correlated with physiology, anatomy and spatial positioning. However, we still use a comparative framing of the problem, as the transcriptomics clearly predicts not only greater within-human neuronal supragranular diversity than described by morphology and physiology, but also substantial species differences in cellular diversity and within-class variation (most notably in the deep layer 3 human types that are not found in layer 2/3 in mouse). We highlight again that the species differences in gene expression and cellular density are observed in multiple cortical areas in both mouse and human. However, we have moved the mouse visual cortex Patch-seq data to extended data with appropriate caveats about interpreting a direct comparison due to an unknown degree of areal variation in morphoelectric properties. This reframes the results closer to the "what is a cell type" theme suggested by Referee #1 but allows inference about the functional implications of cell types simply not present in the supragranular layers of mouse cortex.

Specifically, we have reorganized the manuscript in the following ways:

- 1) Shifted the focus to understanding how the organizing principles of transcriptomically defined cell types relate to the morphoelectric properties of the diverse population of L2-L3 excitatory neurons in the human cortex. This is the strength of the paper and ultimately yields a functional connection to a transcriptomics dataset that is derived using dissociated neurons. We find substantial correspondence between the modalities and that previously described phenotypic heterogeneity can be better understood within the framework

established by the transcriptome. As large-scale transcriptomic datasets are currently leading the way in cell type definitions within and between species, this study is valuable in that it reinforces their relationship to functionally defined types, and that this relationship is robust even in the most challenging experimental conditions in terms of uncontrolled variation presented by human neurosurgical tissues.

- 2) Moved the mouse VISp Patch-seq data to a supporting role as extended data. The reviewers raise a valid point that our data do not distinguish between cross areal and cross species phenotype differences. Neurosurgical resections yield tissue from the MTG, which unfortunately does not have an unequivocal homologous region in mouse. Although we agree that a more extensive study of multiple mouse brain regions would be valuable, such a study is beyond the feasible scope of this manuscript and would ultimately pull the focus away from the human L2 and L3 types. Furthermore, with reduced laboratory access due to COVID-19 it is simply not feasible to perform such an extensive and resource-intensive expansion of the study. We still find tremendous value in comparing the human data to the transcriptomic and phenotypic diversity of mouse L2/3. We have added analysis of additional mouse brain regions to Figure 1, as well as additional literature cited in the introduction to reinforce the relative homogeneity of mouse L2/3 excitatory neurons.
- 3) As part of the refocus to the phenotypic description of the human L2-L3 transcriptomic types, we have included a revised Figure 4 that shows morphoelectric summaries of all 5 human L2-L3 excitatory transcriptomic types. Combining the superficial and deep neuron types in a single figure emphasizes the key finding of this study – the substantial morphoelectric diversity of human L2-L3 excitatory types, and how that diversity can be explained in the context of transcriptomic types. Here we also add an additional analysis of the local axonal extent of the 5 types. The description of the depth dependent phenotypic diversity in the FREM3 type is unchanged. In the final portion of the manuscript, we focus on the 3 neuron types found in deep L3, and how they are distinct from both the superficial neuron types as well as each other.

Referee 1

This is a well-organized and technically strong manuscript as expected from the Allen Institute, which continues to produce superb resources and analyses.

We thank the reviewer for the careful reading of the manuscript and the detailed constructive feedback.

The paper intends to describe the ways in which the superficial layers of the human cortex have expanded during brain evolution compared with the mouse. While the concept of this paper affords an opportunity to do just that – an

unprecedented cross species comparison with single-cell and PatchSeq approaches that provide the ability to match transcriptome, morphology and function in same cells in human tissue from surgery resections, the evolutionary comparison is not convincing and is not integrated with other far more convincing published demonstrations of primate and human brain evolution.

We have reframed the manuscript to focus on the correspondence between morphoelectric and transcriptomic neuron types in human (and in mouse as a comparison), as described above.

Further, the authors do not efficiently compare and establish their work within the current knowledge of the field.

We thank the referee for their direction to better put the study in the context of the prior literature and have reformatted the introduction and conclusion to better cite relevant literature and added comparisons with previous data sets. See comment #9 below for more details.

Even with these large weaknesses, one does have to appreciate that this work addresses another important question – not necessarily anything to do with evolution, but the correspondence between transcriptional diversity and functional diversity at the physiological level as mentioned above should probably be the focus of the paper – how to define a cell type?

We agree with the referee that establishing the correspondence between transcriptomic, physiological and anatomical properties of neurons in the human cortex is the most important advance in the study and have refocused the paper as suggested. Our strategy of using the transcriptome as a guide has worked to map cellular diversity in terms of types and gradients compared to prior studies based on morphology and physiology alone. The manuscript now focuses primarily on the success of this approach to (1) define human cell types using transcriptomics alone, (2) map Patch-seq cells onto these cell types, and then (3) assess physiological and morphological diversity of these types.

Still this requires a far more earnest comparison to multiple mouse regions and other well accepted human single-cell cortical datasets. While it is understood that this is not an easy feat, the field needs a better way to ground the expansion of transcriptionally-defined cells onto regional, anatomical and functional features of neurons into commonly accepted types.

While we acknowledge that it would be desirable to compare Patch-seq results across a

variety of mouse (and human) cortical regions, this is unfortunately well beyond the scope of this manuscript given the resources and time required to do so. Given the referee's concerns about comparing homologous regions, we also feel this would not sufficiently address the issue of a comparison to human MTG. As we have described, we have focused the study largely on human Patch-seq analysis now and moved the mouse Patch-seq (itself a significant contribution) to extended data given the caveats with the interpretation of a comparison.

On the other hand, there are generalizable aspects of cortical organization that justify a comparative framing of the significance of the study in understanding the cellular organization of human cortical organization compared to mouse as the dominant model organism in neuroscience. The cellular diversity specifically in supragranular layers, which is disproportionately expanded in human cortex compared to mouse, is lower in mouse compared to human by any metric in every cortical region. We add to Figure 1 that the variation in cellular density and soma size between human and mouse is common to both visual cortex and the temporal association area (TEa), a region commonly used as a comparator to MTG (despite little to no evidence of homology of these regions across species). We also show that the cellular diversity and variation within and between types in VISp is also very similar in published data from the very different mouse ALM region (mouse TEa data is not available).

Furthermore, we now reference results from a parallel manuscript that is also part of the BRAIN Initiative BICCN project comparing mouse, marmoset and human primary motor cortex (<https://www.biorxiv.org/content/10.1101/2020.03.31.016972v2>). Specifically, in this study we map the MTG data to M1 data and show that the human cellular diversity described here is also seen in human M1. We hope the referee agrees that these data justify a comparative framing of the problem since this issue of supragranular diversity and depth-dependent axes of variation are a generalizable one and important as they illustrate cortical features that cannot be studied in mouse.

Regarding a comparison of the current transcriptomic data to prior datasets, we have in fact already published on such comparisons (see Hodge et al. 2019). An important element of that comparison is that the deep sampling performed showed higher resolution than other published studies in terms of sensitivity (more reads/nucleus, more genes detected) and cellular diversity (more cell types). As a consequence, these studies did not reveal the full complement of cell types and depth-dependent continuous variation that we demonstrate in this study correlates with other cellular phenotypes. This is perhaps an underappreciated aspect of the current study that we now comment on, as the field in general is uncertain whether the increased cell type resolution seen in transcriptomics studies with greater sampling is significant at the finest type versus a higher-order subclass level. Here we show this type resolution is indeed correlated with other cellular phenotypes in a way that would not be seen with coarser sampling and cellular resolution as a guide.

This paper extends previous work on single cell analysis, starting by concentrating on molecular comparisons between human and mouse to identify evolutionarily important differences. This comparison relies on more detailed analysis of data

that has been published in two previous manuscripts that they reanalyze in more detail here. They use data from visual cortex in mouse, which is likely a sample of convenience, as it is not necessarily the most evolved or related to human/primate cognitive traits, and does not match with the human region profiled. Essentially the comparison is between a primary sensory region in one species with an association area in another. In fact, visual cortex is probably the most distinct cortical region in primates and one wonders what a direct comparison of human visual cortex with MTG would show?

We agree this is an interesting question, as the primate visual cortex is by far the most specialized and cytoarchitecturally specialized region. Such a comparison will eventually be part of a future study based on transcriptomics but outside of the scope of the current study.

Furthermore, it is simply impossible at present to obtain neurosurgically resected tissues from human visual cortex for obvious reasons.

It is worth noting that the mouse primary visual cortex is nowhere near as specialized compared to other mouse cortical regions cytoarchitecturally or based on physiological analyses in the literature. Nevertheless, we acknowledge the point that the species differences in morphoelectric properties could be due to a combination of differences between species and between cortical areas. As mentioned above, we have removed the comparison with mouse Patch-seq data and expand our cross-species comparison in Figure 1. We now show that the heterogeneity and distinctness of transcriptomic cell types in mouse ALM are comparable to what we initially reported in VISp. While this analysis does not directly address what human VISp would look like, it shows that at minimum our cross-species results involving human MTG hold when compared to two distant and functionally distinct cortical areas in mouse.

That the mouse shows less regional difference than human in cortex does not remediate this issue as claimed in the introduction. This is quite a thin argument to support comparing vastly different brain regions to understand how the superficial layer of the human cortex evolved. While that statement is true for the magnitude of differential gene expression in bulk RNAseq experiments, where subtle changes in gene expression underlying cellular subtypes are not easily discovered, the simple amount of differential gene expression is not necessarily informative of cellular diversity. Any conclusions based on these comparisons only confuses the already rapidly growing complex landscape of transcriptionally defined cells. Because of these regional differences, the comparison between human and mouse is fraught from the beginning, and only slightly mitigated by comparisons between glut neurons that are transcriptionally matching.

While we feel that the amount of differential gene expression (in combination with

other morphological and transcriptomic results presented in Figure 1) is a good reflection of cellular diversity, this is now presented solely as motivation for performing our within-human cell type analysis. As stated above, the reviewer concerns about cross-species comparisons are well taken and the mouse Patch-seq data is now included as a supporting, extended data in the revised manuscript.

Lastly, the various sizes and thickness differences between species have been generally remarked on before and there is a literature on the size of human and primate cells and their extensive dendritic branching, etc. There is no question that the lamination of mouse cortex and human are markedly different, but so is human visual cortex from MTG.

We acknowledge this point again but emphasize that the expanded supragranular layers in human and nonhuman primate cortex is a generalizable phenomenon in many neocortical areas, and we observe an increase in cellular diversity transcriptomically across multiple areas from both human and mouse. We now also present a comparison of cell density and soma size of human MTG to a putatively better matched mouse region TEa in addition to VISp. We find that our principle findings hold for this region as well, including smaller, more densely packed cells than in human and with similar distribution to those in mouse VISp, suggesting that differences with human specifically in supragranular layers are not due solely to the different brain regions selected in each species. As we demonstrate in human MTG, the transcriptomic cell types can be differentiated with other phenotypes. The description of transcriptomic cell type properties in any mouse cortical region, and demonstration of more minimal phenotypic differences between types and nothing that looks like the novel deep L3 types or strong depth-dependence in L2/3, is a valuable addition that further bolsters the use of the transcriptomics to predict cellular variation within and between species.

The work here also derives narrowly from theories and approaches seemingly based primarily on the authors internal work and lacks scholarly comparison to the field or historically what has been found. The authors downplay the absolutely enormous amount of prior work defining the diversity of glutamatergic cell types across different areas of the mouse cortex – specialized specifically for the mature circuit they function within such as intrinsic membrane properties, morphology, and transcriptomics.

We thank the referee for their strong suggestion to better reference the prior literature on glutamatergic cell diversity and have attempted to better cite this literature. We acknowledge we had perhaps de-emphasized prior studies in favor of beginning with the new guide of transcriptomics. However, although there is indeed a large body of work describing diverse morphoelectric properties of glutamatergic neurons in the mouse cortex, the majority of the variation across areas in these properties (as opposed to

long-range connectivity or behavioral function) has been described in the neuron types that are located in deeper layers. Indeed, previous large-scale morphoelectric studies of cortical neurons have revealed a mostly homogeneous population of L2/3 glutamatergic types within a cortical area (Gouwens et al.

2019, Markram et al., 2015), and also between areas in mouse compared to primate (Gillman et al., 2017). Our mouse VISp Patch-seq data here support this relative homogeneity, and we find that human MTG L2-L3 is different in that we observe a degree of diversity that is not seen in mouse L2/3. We have rewritten the introduction and discussion to emphasize this point and included the relevant citations that show morphoelectric homogeneity in mouse L2/3 glutamatergic neurons, as well as work in the non-human primate, where a similar diversity to human in morphoelectric properties of L2-L3 glutamatergic cells has been described. We would be most receptive to additional information about mouse L2/3 glutamatergic neuron variation in the types of properties measured here. We hope the referee agrees that the removing of the direct comparison of mouse VISp to human MTG in the main manuscript and better citation of prior literature alleviate these concerns.

The authors downplay the diversity that is known in the across the mouse cortex and within excitatory neuron populations. Prior to the explosion of single-cell datasets, decades of work have gone into carefully characterizing cell types in the mouse model by basic criteria of location, intrinsic membrane properties and basic transcription factor cascades or markers selective for those cells (Functional and anatomical correlates: Fishell, Macklis, Arlotta, Marin, Rubenstein, McBain, Barres, Stevens, Krigstein, etc).

Authors do not ground their discovered cell types in the vast literature that which define cells based on these multi-modal features. This is a HUGE loss given they have a potentially powerful and unique dataset composed of nuclear transcriptome sequencing, morphology and intrinsic membrane properties within human cells within the superficial layers cortex.

As mentioned above, while we acknowledge the diversity of excitatory neurons across the mouse cortex, most of that diversity is seen in the deeper layers. Surveys of morphoelectric diversity of mouse L2/3 neurons show a relatively homogeneous population (Gouwens et al. 2019, Markram et al., 2015).

This dataset offers them a very special understanding of neurons at a level mostly unprecedented in the human, which is what I view as the main strength of the manuscript – the extensive anatomical and physiological characterization.

Fundamentally the major contribution of the work is in the several t-type to other phenotype comparisons made – other related strengths of this work include: a)

very thorough characterization of the cells they were able to profile through Patch-Seq and single-cell RNAseq in brain slices from human brain resections. b) development of an efficient scoring method for quantifying the pathology of tissue during analysis to signal healthy state of cells (as tissue comes from surgeries to remove tumors, epileptic centers, etc). c) Interrogating almost 300 glutamatergic neurons within the superficial layers of the human brain (from parietal, frontal, and temporal cortex) for three simultaneous modalities in each cell: nuclei-transcriptome, morphology, and functional membrane properties. This last point marks the key strength of this paper, which perhaps the entire paper should be refocused around: showing this correspondence in humans and relationship between molecular, anatomical/circuit placement and physiological features.

We thank the reviewer for recognizing the unique value of the dataset and suggesting reframing of the paper to focus primarily on the morphoelectric diversity of the human neurons. We agree and have rewritten the manuscript as described above.

The mouse-human part is just not remarkable – For example, a major finding shown in figure 4 is that human L2 and L3 t-types show “greater morphological and electrophysiological differentiation than homologue mouse t-types.” This is interesting, but not unexpected.

We confess that we do not understand the critique that showing greater phenotypic variation between human cell types than mouse cell types is not an interesting result. However, given the removal of the comparison to mouse VISp in the main manuscript this should no longer be a concern.

Again, these comparisons need to be more firmly placed within current knowledge. This could be addressed computationally with existing datasets, but far more substantial comparisons are required for a true understanding of how diversification of the supragranular cortex evolved from mice to human.

For these reasons (and others described elsewhere in the response) we have moved the mouse VISp Patch-seq data and associated comparison to supporting data in the revised version of the manuscript. We emphasize this was already a comparative rather than evolutionary study, and take the referee's point.

On a more minor, but still important level, the authors focus on supragranular neurons and cite several findings that they claim show the importance of these layers in human evolution, including their vulnerability in Alzheimer's disease. This

putative relationship to Alzheimer's Disease is somewhat of a non-sequitur, because it has little to do directly with human brain evolution – are the authors saying that this vulnerability is human specific, and that these pyramidal neurons are not affected in mice? -- because that has not been demonstrated. From this perspective, the Von Economo neuron would be more important to look at, which they do not.

In the context of the mouse-human comparison that was central in the previous version of the manuscript, we can see the reviewer's point that the Alzheimer's disease (AD) connection was not directly related. However, in the reframing of the manuscript solely focused on the human excitatory cell types, the AD connection becomes even more relevant, as we reveal that a neuron population previously shown to be selectively vulnerable in a neurodegenerative disease specific to human is not present in mouse. We provide evidence that a specific subset of large pyramidal neurons in deep layer three (deep *FREM3* and *CARM1P1* cells but not *COL22A1* cells) are labeled with the SMI-32 antibody and also express *NEFH* (the gene coding for the neurofilament protein labeled by SMI-32). These results, in combination with previous work from one of the co-authors (Patrick Hof) showing loss of SMI-32-immunoreactive neurons in postmortem AD brains, strongly suggest a human-specific vulnerability (Bussi re et al., 2003; Hof and Morrison 1991). We believe this is a very important result from the perspective of transcriptomically distinguishable cell types showing selective vulnerability in disease as these methods are on the cusp of large-scale application to AD and other diseases by ourselves and many others. That being said, we have now added a discussion point mentioning that further cell typing studies on AD tissue would prove whether these are the only cell types lost in AD or whether they are only a subset of cell types lost in AD.

Further, the expansion of the cerebral cortex involves increased cell division and the expansion of the outer subventricular zone in primates and humans. There are specific regions of the genomes (regulatory) and genes (targets) that have been associated with the expansion of these cell classes in several publications and none of these are cited or discussed.

This is a very interesting question but one that we feel is outside of the scope of the current study focused on the morphoelectric correspondence to transcriptomic cell types, particularly given the removal of the direct comparison to mouse. In addition, our study is focused on the mature adult human brain and it would be entirely speculation to discuss development.

How does the diversity of the cell types identified relate to identified sequence differences in regulatory regions or the genes that they regulate in the human genome? Again, the focus on these layers is well deserved based on their importance in human brain function and evolution, but some of the rationale and the actual human mouse comparison does not work well, and is not convincing.

We also find these are intriguing lines of research but feel they are outside the scope of this specific study and would be better approached by adding single cell epigenomic analysis to the current transcriptomic datasets.

Reviewer 2

The manuscript "Human cortical expansion involves diversification and specialization of supragranular intratelencephalic-projecting neurons" by Berg...Lein contains many original and impressive results.

They include at least:

a) Stereotypy of neuron types, assessed by transcriptomic, morphological and electrophysiological combined approaches across several human neurosurgical specimens that are heterogeneous for many parameters such as age, gender, ethnicity, disease's condition and severity.

b) Demonstration that transcriptomes of neurons in human supragranular cortex are well-correlated with their morphological and physiological features, as well as cortical depth.

c) Identification of five novel neuron t-(transcriptional) neuron types in the human supragranular cortex. Specifically, LTK t-type that contains small neurons and is largely restricted to L2; GLP2R t-type that is found in L2 and superficial L3 with few dendrites and minimal branching in L2; FREM3 t-type that is found in L2 and L3, and shows continuous transcriptomic, morphological and physiological variation as function of cortical depth; CARM1P1 and COL22A1 t-types that are detected exclusively in deep L3, they have wide, highly branched apical dendrites and tall, very sparse apical dendrites, respectively.

d) Observation that deeper L3 FREM3 and CARM1P1 neurons are labelled by an SMI-32 antibody for subunits of neurofilament proteins. Notably this SMI-32 immunolabeling is lacking in Alzheimer's disease (AD), suggesting that AD progression could target specific cortical neuron types in humans (FREM and CARM1P1).

Overall, the results contained in this manuscript, represent a significant advancement in the understanding of human cortical glutamatergic neuron types. The novel information submitted, when it will be published, is likely to have a strong influence for future research in the field. As a whole, the data submitted also provide a remarkable demonstration that transcriptomic is a powerful approach that can be successfully used in combination with structural and functional approaches. The manuscript also provides an important, unprecedented

and comprehensive quantitative analysis of neuronal properties in relation to cellular markers of pathology in the cortical samples taken from neurological patients.

We thank the reviewer for the careful reading of the manuscript and constructive feedback. We observe the above description of strengths is focused on the novel human dataset (similar to the primary feedback of referee #1). We have rewritten the manuscript to focus on the phenotypic diversity of the human L2 and L3 excitatory neuron types and how that diversity can be explained using transcriptomic types, as described above.

I have the following constructive criticism.

1) A comparison is made between neurons of the human middle temporal gyrus (MTG) and mouse primary visual cortex (VISp). However, there is strong specialization of principal neurons according to cortical functional areas, which was also shown by one of the studies showing single cell data in the mouse (Tasic et al. 2018), where subtypes of principal neurons were almost exclusive either for lateral motor or primary visual cortex. Therefore, the choice of comparing L2/3 neuron types in the mouse visual cortex versus human MTG is sub-optimal. To compare homologous areas in the human and mouse cortex would improve this situation and is highly recommended for a new edition of the manuscript.

While we agree that a comparison of homologous regions would be preferable, unfortunately this is not feasible to include in the present study. First, the homologous region to MTG is rather unclear, and it is not possible to obtain sufficient tissue from human cases from other cortical regions. These regions never include the main primary sensory and motor areas studied in mouse. In mouse the temporal association area (TEa) is sometimes used as a closer comparison. We now include histological analysis of TEa in addition to VISp in Figure 1 but adding a completely new study of Patch-seq in mouse TEa is an enormous project that is not feasible to add at this time. Given this issue was raised by both referees, we have instead opted to take referee #1's suggestion to focus the study on human Patch-seq and move the mouse VISp Patch-seq data to extended data. We still include the mouse/human comparison in Figure 1 based on histology and transcriptomics to establish the human results in the context of increased diversity seen generically in human versus mouse cortex however, and there add the comparison to TEa.

Specifically, we have:

- Moved the mouse Patch-seq data and associated analyses to supporting information and focused the manuscript on characterizing human supragranular glutamatergic types.
- We have expanded the initial comparison of human and mouse cell types based on gene expression and histology in Figure 1 to include an additional mouse brain region and present this as motivation for studying human rather than as an interesting biological result in and of itself.
- More specifically, we compare gene expression in both mouse ALM and VISp to

human MTG and find that the same patterns hold.

- In addition, we compare histology of mouse TEa (a closer analog to human MTG than VISp) and find that these cells are slightly smaller and packed at a higher density than in mouse VISp, but that all of the comparisons between human MTG and mouse VISp also hold for mouse TEa.

Together we feel that these changes address the valid concerns brought up by both reviewers related to cross-species analyses.

2) Electrophysiological characterization relies on several passive and active parameters that have been carefully collected and analyzed. However, spiking in response to squared depolarizing current pulses lasting several hundred milliseconds represent poorly defined physiological stimuli. They may occur "in real life" for some cells during cortical up-states and desynchronized brain states, but otherwise they are a quite artificial way (although widely used since the 50s - Granit's intracellular recordings of spinal motoneurons) to assess neuron's excitability. The manuscript should acknowledge the relatively poor physiological meaning of this protocol.

We acknowledge that there are many additional physiological measurements that could be used to better probe specific cell type function. We revised the manuscript to stress that the intent of the stimulus paradigm is not to probe physiological inputs, but to use a single standardized protocol that can be used to generate electrophysiological features that can be used to compare diverse cell types (and from different species) directly when the identity of each probed cell is not known at the time of measurement.

The study would greatly benefit from an analysis of the synaptic inputs of the transcriptomically-defined neuron types. Due to the differential laminar distribution, shape and complexity of their dendrites, do these neurons receive similar amount of excitation and/or inhibition? What are the plastic properties of excitation and inhibition impinging on these neuron types? I am well aware that a single manuscript cannot address all the relevant questions. However, I do think that to submit at least some of synaptic data here, would make this manuscript more meaningful, stronger and interesting. Also, detailed information on dendritic patterns for each neuron type is given, but no attempts are made to discuss what inputs (cortical, sub-cortical etc) are likely to connect to the various neuron types. A discussion on this should be added.

We agree that this is an interesting avenue of investigation as a follow-up to the basic characterization of the different t-types. However, this is technically an extremely challenging experiment to molecularly identify both pre- and postsynaptic neurons that are not identified prospectively. Effectively this requires establishing the combination of paired or multiple patchclamp physiology and Patch-seq, or multi-Patch-seq. Given the relatively high fail rate for any given data modality with Patch-seq, the success rate with such a multi-Patch-seq approach will be very low and difficult to approach in human tissues in view of significant access limitations. We are trying to develop ways to make

this an efficient strategy, as well as prospective viral labeling of specific neuron types in a slice culture paradigm, but these are in development and unfortunately outside of the feasible scope of the current study.

3) It is a bit surprising that the manuscript does not provide any information on the axons of the morphologically reconstructed neurons. I understand that the axon is cut by the slicing procedure, yet some local axon collaterals should be detected within the slices. To include information on axon's aspect, branches, layers occupied and possibly targets etc, would represent a significant improvement. Some discussion on the possible targets of the newly-defined neuron types should be also added.

The challenge with axon representation in these human slice preparations is that they are very often highly truncated as they span a large volume and exit the plane of the slice in non-reproducible ways. Nevertheless, we have obtained many well labeled axons and are happy to add this information to the manuscript in Extended Data Figure 7. The truncation issues make quantitative analysis of axon features quite challenging beyond the layer containing the soma and the adjacent layers that are typically better preserved. Nevertheless, we now provide a gallery of the recovered axons of high-quality Patch-seq neurons. We also quantify their average axon length histograms, which illustrate the laminar axon distribution pattern of each transcriptomic type and provide insights into the putative post-synaptic targets of each t-type, as well as an additional validation of the putative projection target subclass that each t-type is predicted to belong to. We also show quantifications for several features that differ significantly between types, despite the fact that each type has a similar amount of recovered axon

4) Data shown in Fig 2f. What could be the meaning of the more hyperpolarized membrane potential in the high IBA1 group (more pathological) compared to the other groups (less pathological, normal)?

We thank the referee for suggesting we take a closer look at the apparent difference in membrane potential variation as a function of IBA1 expression (presumably reflecting inflammation). It turns out that this effect is not statistically significant. Rather, membrane potential varies strongly as a function of cortical depth. The small number of cells in the high IBA1 group was imbalanced compared to the larger population in terms of the depth distribution. When depth is considered using regression analysis, the phenotype of cells belonging to the high IBA1 group is not significantly different from the low IBA1 group. We have revised the text accordingly.

5) The introduction could be improved. It mostly repeats the main results of the paper. Instead, it should provide more information on the state of the art of the field, background information and explain the rationale behind the methodologies used.

We have updated the introduction extensively to address this point, as well as to improve citations of relevant literature as suggested by both reviewers.

6) While the number of human cells for Patch-seq is rather high, the number of mouse cells studied is rather low, e.g. in Fig. 3c right panel <10 cells align to Adamts2 cluster. The Authors should increase the number of mouse Patch-seq neurons to provide a more robust comparison with human data.

As described above we have decided to move the mouse data to extended data in response to issues of comparing data from different cortical regions across species. Furthermore, due to constraints imposed by the current health crisis, we are not able to improve so extensively the size of our mouse data set in the reasonably near future. We hope the referee agrees that given these circumstances and reframing of the study this is less of an issue.

7) Figure 1 data. Based on UMAP plot and panel e, FREM3 neurons might be further separated in t-subtypes. Did the Authors try to find markers that potentially separate FREM3 neurons in sub-clusters? Could be that 5 subtypes in MTG is an underestimation due to the relatively small number of cells analyzed?

This is an excellent observation that illustrates a common issue with clustering in single-cell RNA-seq studies. For a given cell type, how should we decide whether the clustering is under-split, sufficiently split, or over-split? Ultimately, we select a specific computational method and then try to substantiate whether those results are meaningful using orthogonal datasets such as the Patch-seq method used here. In Hodge et al. 2019, we chose to keep FREM3 as a single cluster because the high amount of variation in this cluster is continuous variability (e.g., a gradient with no marker genes for intermediate parts of the gradient) rather than clustering further to split that gradient. We show in Figure 1d of this manuscript (and in supplemental materials from Hodge et al. 2019) that FREM3 can be split into finer types. We used the lower degree of clustering (FREM3 as a single cluster) as the reference for the current study, and find that there are aspects of these other phenotypes that mirror the continuous variation and others that are more consistent with further splitting of deep versus superficial types. For example, many physiological and morphological characteristics vary smoothly as a function of depth. On the other hand, many genes are expressed selectively in the deeper portion of the FREM3 gradient, including the SMI-32 labeling that has been linked with long-range ipsilateral projecting neurons in macaque monkeys and selectively depleted in Alzheimer's disease. It would seem that there is validity both in describing this type as a single type with graded properties, and also as a superset that can be meaningfully split. Similar results have been reported recently for continuous versus discrete characteristics in mouse for excitatory (Scala et al., 2020) and inhibitory (Gouwens et al., 2020) neurons.

8) Figure 3, Extended Figure 2. Representation of neurons in Patch-seq data should be shown per sample. While it is certainly difficult to label that many human samples at UMAP plots, such information could be provided as a table,

where for each human sample the number of neurons for Patch-seq is given for each subtype. In order to be more confident that each subtype was found in many human samples, the number of human samples where a subtype was found should be indicated.

This is a great point, which is now addressed in the manuscript in two ways. First, we include additional panels in Extended Data Fig. 2 to show each cell color-coded by donor in both the transcriptomics and electrophysiology UMAPs. Second, we present a table showing the number of donors represented in each cell type. In both cases, the results show no obvious donor bias, with cells from each donor scattered across multiple types. However, we note that there is high variability in the number of Patch-seq neurons per donor, ranging from 1 to dozens, making it challenging to perform any high confidence statistical test to this effect. Here is an example plot in the transcriptomics UMAP presented in a slightly different way from the manuscript for clarity.

9) Figure 4, Extended Figure 7. Integration of mouse and human data is obviously a difficult task and from the results and methods submitted in this manuscript it is not completely clear how much effort has been spent for a robust integration of inter-species data. Did the Authors try to change data integration parameters by reducing some differences between mouse and human cells allowing for better inter-species alignment? The Authors used Seurat for data integration, however there are a number of other tools that performed well in comparative studies, did the Authors try some other tools?

This point is well taken, and we have spent a great deal of effort comparing different data integration methods both between species and between data types. In Hodge et al. 2019, we compared two separate strategies for cross-species integration (Seurat and scAlign) and found the same results overall. However, we have removed the cross-species component of the present study and we do not present any such comparisons in the manuscript.

We have also tested multiple variations of three different mapping strategies: (1) Seurat (as described in this manuscript), (2) a variation of the tree-mapping strategy presented in Gouwens et al. 2020 for analysis of GABAergic cell types in mouse Patch-seq and (3) a

correlation-based strategy comparing expression of marker genes in Patch-seq vs. cluster medians of the scRNA-seq data. In all cases, we find between 75-95% of cells mapping to the same cell type when comparing pairs of methods. While this number is lower than one might hope for it is consistent with similar analyses performed using dissociated FACS sorted cells and therefore does not reflect a quality issue with Patch-seq data per se, but rather the fact that celltype definitions are not totally discrete, and single cell measurements are not totally accurate (dropouts, etc.). As such, many cells show expression patterns that are not unambiguously mapped between highly similar transcriptomic cell types, although how much of this is biological and how much technical is difficult to assess. Finally, and most importantly, despite the imperfect agreement of specific cells, the statistical results regarding morphology and electrophysiology for each cell type remain relatively unchanged regardless of the mapping method used.

10) Extended Figure 2, age panel. It is surprising to see lack of differences between neurons from 20 and 80-year-old brains. The Authors should comment this result in the text, and provide an interpretation for the lack of aged-related differences in the parameters studied.

We explored the possibility of age effects more quantitatively using a feature-wise regression analysis, in addition to the dataset-level visualization shown in the Extended Data. The results were suggestive but not conclusive of a possible age effect on two features, rheobase and f-I slope; these relationships are significant at FDR-corrected $p < 0.05$ only if the regression included cell-type identity of the cells as additional factor. No age effects on morphology features were significant, but due to the smaller sample size effects of a comparable size to the electrophysiology would likely be missed. Nevertheless, it is highly notable that we did not observe obvious age-related signatures and we now comment on this in the manuscript.

11) In the discussion, the manuscript should attempt to better correlate the novel glutamatergic neuron types proposed by using transcriptomics, functional and anatomical analyses with the already known glutamatergic neurons of the supragranular cortex found in humans or other species, such as primates, cats or rodents.

We have updated the introduction extensively to address this point, as well as to improve citations of relevant literature as suggested by both reviewers.

References

Bussi re, T. *et al.*. Progressive degeneration of nonphosphorylated neurofilament protein- enriched pyramidal neurons predicts cognitive impairment in Alzheimers disease: Stereological analysis of prefrontal cortex area 9. *Journal of Comparative Neurology* **463**, 281–302 (2003).

Gilman, J. P., Medalla, M. & Luebke, J. I. Area-Specific Features of Pyramidal Neurons—a Comparative Study in Mouse and Rhesus Monkey. *Cereb Cortex* **27**,

2078–2094 (2017).

Gouwens, N. W. *et al.*. Classification of electrophysiological and morphological neuron types in the mouse visual cortex. *Nat Neurosci* **22**, 1182–1195 (2019).

Gouwens, N. W. *et al.*. Toward an integrated classification of neuronal cell types: morphoelectric and transcriptomic characterization of individual GABAergic cortical neurons. (2020) doi:10.1101/2020.02.03.932244.

Hodge, R. D. *et al.*. Conserved cell types with divergent features in human versus mouse cortex. *Nature* **573**, 61–68 (2019).

Hof, P. R., Cox, K. & Morrison, J. H. Quantitative analysis of a vulnerable subset of pyramidal neurons in Alzheimer's disease: I. Superior frontal and inferior temporal cortex. *J Comp Neurol* **301**, 44–54 (1990).

Markram, H. *et al.*. Reconstruction and Simulation of Neocortical Microcircuitry. *Cell* **163**, 456–92 (2015).

Scala, F. *et al.*. Phenotypic variation within and across transcriptomic cell types in mouse motor cortex. (2020) doi:10.1101/2020.02.03.929158.

Reviewer Reports on the First Revision:

Referees' comments:

Referee #1:

The authors submit a revision of the previous manuscript, which was a very strong initial submission, that also had a few serious weaknesses, especially in terms of emphasis. They address the biggest issue raised across reviewers – trying to reach generalizable evolutionary conclusions from detailed comparison of a different cortical region from each species – not by garnering more molecular data from another mouse region in detail, but by reframing the paper somewhat. Given the amount of data and effort represented in this work that is a very reasonable compromise, and the work is much easier to digest in this current form. In fact its main messages are far more clear and convincing.

The original paper confounded what could be differences between brain regions with species differences by comparing human MTG to mouse VC at the single cell level. That was certainly not the optimal study design for cross species comparison, since it is known that human visual cortex is highly divergent from other cortical regions, and the data presented in mouse was not strong enough to support VC as a representative cortical region from mouse. Contrary to their claims, without the data, the comparative framework is not that strong. That does not preclude the paper from publication – it is extremely well done and represents beautiful work pertaining to key issues in cell type definition (e.g. Figure 4), which they do across species. In fact, the combination of anatomy transcriptomics and physiology defining human MTG cells is fantastic. But, the original comparative evolutionary framing about supragranular layers is not a strength, especially given the unique qualities of these layers in VC. Overall, in their reframing of the work more focused on the relationship of different levels of analysis (transcriptomics, physiology etc.) and the diversity of cell types in human layers II/III and a parallel analysis in mouse VC they have done a good job in responding the critiques. Further, the demonstration that differences in physiological properties of the cells are not related to obvious pathology (gliosis, inflammation), and comparison of patch-seq to nucSeq are very important and sets this work apart. This manuscript represents another

compilation of outstanding work from the Allen Institute that carefully demonstrates the relationship between transcriptome diversity and morphometric and physiological features, and that the transcriptome diversity does represent key elements of functional diversity. In this regard it is a tour de force.

A few minor issues to consider:

The work starts by comparing human MTG, which is a surgical specimen to two mouse regions, TEa and primary VC on a morphological level in terms of cytoarchitecture and cell density. This and other published data is used to support the notion that supragranular cortex in mouse is so similar across mouse that it does not make a difference which region is used to compare with human. The anatomical-only comparison is not sufficient, and detracts from the overall logic of the paper, which is the power of single cell RNA analysis to distinguish between cells that otherwise might appear the same, microscopically or using standard immunocytochemistry. Logically, the authors can't have it both ways – either the molecular profiles are crucial for cell type diversity, or anatomical/cytological methods are sufficient. Do they know that mouse VC has essentially the same superficial layer composition as association areas at the molecular level? Further, that supragranular cortex is expanded across the cortex in primates in humans does not address the main issue – in fact this expansion is likely a further driver of regional diversity.

The sentiment that preceding work defining the developmental origins and regulatory changes in the genome that underlie the expansion of these layers, and pointing to their expansion as a major feature of NHP and human brain evolution, is not worth citing in the Introduction is odd; this previous work provides substantial framing for some of the questions being addressed. Because different methods are used does not obviate their relevance to each other. For example, it is easy to assess whether some of the genes in the human specific cells are under control of human specific or evolved enhancers. That would provide strong evidence for their findings being broadly relevant to evolution of the brain.

Referee #2:

I acknowledge that the Authors have resubmitted a stronger version of their manuscript and have carefully considered all my previous criticism.

Specifically:

1) Comparison between neurons of the human middle temporal gyrus (MTG) and mouse primary visual cortex (VISp). The addition of new data of Patch-seq in mouse TEa – and not only histology - would have been the best option. This would have required a significant amount of additional work, yet this would have been feasible due to human and technical resources available by the consortium of the authors. Instead, the revised manuscript (R1) focuses on human Patch-seq and move the mouse VISp Patch-seq data to extended data. In my opinion, this choice is sub-optimal and misses an opportunity, but it is an acceptable improvement.

2a) Spiking evoked by depolarizing current pulses. I thank the authors for having stressed in R1 the poor physiological significance of evoking neuron firing with squared depolarizing current pulses of artificial intensity and duration.

2b) Evaluation of the synaptic inputs of the transcriptomically-defined neuron types. I acknowledge that to include an analysis of this type of data in R1 can be beyond the main aim for the submitted work. As the authors suggested, I am also looking forward to seeing this type of analysis performed in viral labeling of specific neuron types in slice cultures in the near future.

3) Analysis of the axon of the morphologically reconstructed neurons. R1 includes a gallery of the

recovered axons of high-quality Patch-seq neurons in Extended data figure 4 (not 7). The axonal reconstructions are very partial, due to the technical difficulties, and provide limited information that however enriches the manuscript.

4) Analysis of the membrane potential of the recorded neurons. I am glad that this comment helped to revise the conclusion reached by this set of data.

5) Introduction. In R1 this section of the manuscript is improved.

6) Number of Patch seq data in the mouse. I agree that since R1 focuses more on human data it become less important to enhance the number of observations in the mouse.

7) Clustering analysis. The authors raised some interesting considerations to my comment in their rebuttal. They should add some comment text on this issue in the manuscript too (probably methods or discussion should be the best places).

8) Representation of neurons in Patch-seq data should be shown per sample. R1 addresses this issue by including additional panels in Extended Data Fig. 2 to show each cell color-coded by donor in both the transcriptomics and electrophysiology UMAPs and a table showing the number of donors represented in each cell type. This additional information significantly improves the manuscript.

9) Data integration methods. Same as #7: they should add their interesting considerations to my comment in the manuscript too.

10) Age data. I acknowledge that the Authors further explored this issue and commented it in R1.

11) Discussion. I am glad that the Authors revised and improved the discussion in R1.

Author Rebuttals to First Revision:

Referees' comments:

Referee #1 (Remarks to the Author):

The authors submit a revision of the previous manuscript, which was a very strong initial submission, that also had a few serious weaknesses, especially in terms of emphasis. They address the biggest issue raised across reviewers –trying to reach generalizable evolutionary conclusions from detailed comparison of a different cortical region from each species – not by garnering more molecular data from another mouse region in detail, but by reframing the paper somewhat. Given the amount of data and effort represented in this work that is a very reasonable compromise, and the work is much easier to digest in this current form. In fact its main messages are far more clear and convincing.

The original paper confounded what could be differences between brain regions with species differences by comparing human MTG to mouse VC at the single cell level. That was certainly not the optimal study design for cross species comparison, since it is known that human visual cortex is highly divergent from other cortical regions, and the data presented in mouse was not strong enough to support VC as a representative cortical region from mouse. Contrary to their claims, without the data, the comparative framework is not that strong. That does not preclude the paper from publication –it is extremely well done and represents beautiful work pertaining to key issues in cell type definition (e.g. Figure 4), which they do across species. In fact, the combination of anatomy transcriptomics and physiology defining human MTG cells is fantastic. But, the original comparative evolutionary framing about supragranular layers is not a strength, especially given the unique qualities of

these layers in VC. Overall, in their reframing of the work more focused on the relationship of different levels of analysis (transcriptomics, physiology etc.) and the diversity of cell types in human layers II/III and a parallel analysis in mouse VC they have done a good job in responding the critiques. Further, the demonstration that differences in physiological properties of the cells are not related to obvious pathology (gliosis, inflammation), and comparison of patch-seq to nucSeq are very important and sets this work apart. This manuscript represents another compilation of outstanding work from the Allen Institute that carefully demonstrates the relationship between transcriptome diversity and morphometric and physiological features, and that the transcriptome diversity does represent key elements of functional diversity. In this regard it is a tour de force.

We thank Referee #1 for the original constructive comments that improved the manuscript.

A few minor issues to consider:

The work starts by comparing human MTG, which is a surgical specimen to two mouse regions, TEa and primary VC on a morphological level in terms of cytoarchitecture and cell density. This and other published data is used to support the notion that supragranular cortex in mouse is so similar across mouse that it does not make a difference which region is used to compare with human. The anatomical-only comparison is not sufficient, and detracts from the overall logic of the paper, which is the power of single cell RNA analysis to distinguish between cells that otherwise might appear the same, microscopically or using standard immunocytochemistry. Logically, the authors can't have it both ways – either the molecular profiles are crucial for cell type diversity, or anatomical/cytological methods are sufficient. Do they know that mouse VC has essentially the same superficial layer composition as association areas at the molecular level? Further, that supragranular cortex is expanded across the cortex in primates in humans does not address the main issue – in fact this expansion is likely a further driver of regional diversity.

We recognize that an anatomical-only comparison of the human samples with two different mouse regions would be insufficient to show that differences are most likely inter-species versus inter-region. We use transcriptomics to compare human MTG to mouse VISp and ALM to further support this conclusion. We detail how the mouse ALM shows similar gene expression complexity to the mouse VISp, with 3 types that form a continuum in transcriptomic UMAP space.

The sentiment that preceding work defining the developmental origins and regulatory changes in the genome that underlie the expansion of these layers, and pointing to their expansion as a major feature of NHP and human brain evolution, is not worth citing in the Introduction is odd; this previous work provides substantial framing for some of the questions being addressed. Because different methods are used does not obviate their relevance to each other. For example, it is easy to assess whether some of the genes in the human specific cells are under control of human specific or evolved enhancers. That would provide strong evidence for their findings being broadly relevant to evolution of the brain.

We thank the referee for the suggestion and have referenced the preceding work as suggested.

Referee #2 (Remarks to the Author):

I acknowledge that the Authors have resubmitted a stronger version of their manuscript and have carefully considered all my previous criticism.

Specifically:

1) Comparison between neurons of the human middle temporal gyrus (MTG) and mouse primary visual cortex (VISp). The addition of new data of Patch-seq in mouse TEa – and not only histology - would have been the best option. This would have required a significant amount of additional work, yet this would have been feasible due to human and technical resources available by the consortium of the authors. Instead, the revised manuscript (R1) focuses on human Patch-seq and move the mouse VISp Patch-seq data to extended data. In my opinion, this choice is sub-optimal and misses an opportunity, but it is an acceptable improvement.

2a) Spiking evoked by depolarizing current pulses. I thank the authors for having stressed in R1 the poor physiological significance of evoking neuron firing with squared depolarizing current pulses of artificial intensity and duration.

2b) Evaluation of the synaptic inputs of the transcriptomically-defined neuron types. I acknowledge that to include an analysis of this type of data in R1 can be beyond the main aim for the submitted work. As the authors suggested, I am also looking forward to seeing this type of analysis performed in viral labeling of specific neuron types in slice cultures in the near future.

3) Analysis of the axon of the morphologically reconstructed neurons. R1 includes a gallery of the recovered axons of high-quality Patch-seq neurons in Extended data figure 4 (not 7). The axonal reconstructions are very partial, due to the technical difficulties, and provide limited information that however enriches the manuscript.

4) Analysis of the membrane potential of the recorded neurons. I am glad that this comment helped to revise the conclusion reached by this set of data.

5) Introduction. In R1 this section of the manuscript is improved.

6) Number of Patch seq data in the mouse. I agree that since R1 focuses more on human data it become less important to enhance the number of observations in the mouse.

7) Clustering analysis. The authors raised some interesting considerations to my comment in their rebuttal. They should add some comment text on this issue in the manuscript too (probably methods or discussion should be the best places).

8) Representation of neurons in Patch-seq data should be shown per sample. R1 addresses this issue by including additional panels in Extended Data Fig. 2 to show each cell color-coded by donor in both the transcriptomics and electrophysiology UMAPs and a table showing the number of donors represented in each cell type. This additional information significantly improves the manuscript.

9) Data integration methods. Same as #7: they should add their interesting considerations to my comment in the manuscript too.

10) Age data. I acknowledge that the Authors further explored this issue and commented it in R1.

11) Discussion. I am glad that the Authors revised and improved the discussion in R1.

We thank Referee #2 for the original constructive comments that improved the manuscript, and for recognizing those improvements.

With regards to 7) specifically, we have added an additional paragraph describing the considerations we used when determining the optimal clustering criteria to the methods (under the heading 'Identifying transcriptomic types').

With regards to 9) specifically, we note that the comparison of mapping strategies that we describe in the rebuttal is detailed in the manuscript methods section, under the heading 'Identifying transcriptomic types'